# Adsorption of Chromium, Copper, Lead, Selenium, and Zinc ions into ecofriendly synthesized magnetic iron nanoparticles

Mutairah S. Al Shammari[1], Hussein M. Ahmed[2], Fatehy M. Abdel-Haleem[3], Nowarah J. Almutlq[1], Mohamed A. El-Khateeb[4] *

1 Department of Chemistry, College of Science, Jouf University, Sakaka, Saudi Arabia, 2 Housing and Building Research Center (HBRC), Sanitary and Environmental Institute, Dokki, Giza, Egypt, 3 Cairo University Centre for Hazard Mitigation and Environmental Studies and Research, CHMESR, Cairo University, Giza, Egypt, 4 Water Pollution Research Department, National Research Centre, Dokki, Cairo, Egypt

* elkhateebcairo@yahoo.com

**Data Availability Statement:** All the data are presented within the article.

## Abstract

The iron nanoparticles (Fe-NPs) have been synthesized using an environmentally friendly and simple green synthesis method. This study aims to obtain an aqueous extract from natural material wastes for synthesizing Fe-NPs. The produced Fe-NPs were evaluated as adsorbents for removing Pb, Se, Cu, Zn, and Cr from aqueous solutions. The formation of Fe-NPs was observed on exposure of the aqueous extract to the ferrous chloride and ferric chloride solutions. The characterization of the synthesized Fe-NPs was carried out using different instrumental techniques. As a function of the initial metal ion concentration, contact time, and various doses, the removal of the heavy metal ions was investigated. The UV-Vis spectrum of Fe-NPs showed a peak at 386 nm, 386 nm, 400 nm, 420 nm, 210 nm, 215 nm, and 272 nm of banana, pomegranate, opuntia, orange, potato, and onion, respectively. The FT-IR spectra confirmed the attachment of bioactive molecules from plants on the Fe-NPs surface. The effective reduction of metal ions was greatly aided by the -OH functional groups. The functional groups were examined and responsible for adsorption process by nanoparticle powder sample, these peaks are 3400 $cm^{-1}$, 2900 $cm^{-1}$, 1600 $cm^{-1}$, 1000 $cm^{-1}$, and 1550 $cm^{-1}$. The magnetization measurements revealed superparamagnetic behavior in the produced iron oxide nanoparticles. Heavy metal ions uptake followed a time, dose, and initial concentration-dependent profile, with maximum removal efficiency at 45 min, 0.4 g, and 3.0 mg/L of metal concentration, respectively.

## 1. Introduction

Maintaining the water quality criteria established by applicable rules and removing dangerous contaminants from wastewater before releasing them into the environment is of utmost importance [1, 2]. Heavy metal discharge-related water pollution is still a major global concern. The massive increase in heavy metal use over the last few decades has inevitably increased

**Funding:** This work was carried out within a research project (No. 223202) financed by the Deputyship for Research & Innovation, Ministry of Education in Saudi Arabia.

**Competing interests:** No conflicts of interest exist, according to the authors, with the publishing of this paper.

the flux of metallic substances in the aquatic environment [3]. In marine, groundwater, industrial, and even poorly treated effluents, heavy metals are significant contaminants. The main causes of the rise in metals discharged into the environment are mining operations, agricultural runoff, and home and industrial effluents [4]. Because heavy metals cannot biodegrade, their harmful effects are exacerbated, making heavy metal pollution a massive environmental issue. Moreover, some of these heavy metals are known to assault the body's active enzyme sites, blocking the enzymes [3–6]. Through the food chain, heavy metals are extremely harmful to humans and have the potential to disguise long-term effects. Thus, it is crucial to remove heavy metals from effluent [5, 7–10].

Precipitation and ion exchange are efficient and cost-effective processes that have been found to be superior to other methods for treating aqueous effluents. Much emphasis has recently been paid to the use of nanoparticles in solid-phase extraction, considering the numerous benefits resulting from their unique properties [11]. The nanoparticles' diameters range from 1 to 100 nm. The properties of the nanoparticle can be significantly altered by surface modification [11–13]. The preparation of iron nanoparticles has attracted both fundamental and practical interest because of their interesting electronic, magnetic, catalytic, chemical, or biological properties [13, 14]. Magnetic iron nanoparticles have many distinct magnetic properties, including the ability to control their size, morphology, and surface properties [13]. Different applications of magnetic nanoparticles exist, such as cancer diagnosis, drug delivery vehicles, and water remediation [15, 16]. Iron nanoparticles (Fe-NPs) can be created using a variety of techniques, including hydrothermal precipitation, sol-gel, emulsion, mono-chemical processing, precipitation, and thermal plasma arc processes [17]. The Fe-NPs produced by physical and chemical processes are complicated, outdated, expensive, and produce hazardous toxic waste that is harmful to the environment as well as human health [18].

The green method is a better alternative to physical and chemical methods for the production of Fe-NPs. It is not only inexpensive, but also less complicated and time-consuming, safe, eco-friendly, and non-toxic [19]. Furthermore, it has a far reduced energy demand, less input waste, and more useful chemical and reagent control. The fact that this is a bottom-up strategy is another benefit [16, 17].

Plant extracts that act like powerful antioxidant compounds such as amino acids, polyphenols, nitrogenous bases, and reducing sugars. These compounds act as covering and reducing agents for nanoparticle synthesis. Because of the diversity of plants, it is possible to control the desired shape and size of nanoparticles by varying the source of the extract. A plant leaf extract used for NP synthesis can be scaled up and applied for larger-scale production in addition to its economic advantages. The metal and metal oxide NPs produced from a plant extract are usually stable even after a month and do not show any visible changes [16].

In this study, iron nanoparticles (Fe-NPs) are prepared based on the green synthesis method, in which extracts from various wastes, such as banana, pomegranate, opuntia, orange, potato, and onion peels are used for the synthesis of iron nanoparticles (Fe-NPs). The Fe-NPs were characterized using different techniques such as x-ray diffraction (XRD), Fourier transforms infrared spectroscopy (FT-IR), visible ultraviolet spectroscopy (UV-Vis spectrum), energy dispersive x-ray (EDX), and x-ray fluorescence (XRF). The particle size, magnetic properties, and morphology of Fe-NPs depend on the conditions of the materials. Heavy metals like Pb, Se, Cr, Cu, and Zn were removed using these produced Fe-NPs. Different factors such as contact time (0, 15, 30, 45, and 60) min, initial concentration of metal ions (1.0, 2.0, 3.0, 5.0, 10) mg/L, and dose (0.1, 0.3, 0.4, and 0.5 g) were applied to determine the efficiency of the nanoparticles.

## 2. Materials and methods

### 2.1 Chemicals

Iron (III) chloride ($FeCl_3.6H_2O$) (98.00%), iron (II) chloride $FeCl_2.4H_2O$ (99.00%), hydrochloric acid HCl (36.00%), and sodium hydroxide NaOH (98.00%) were purchased from Merck (Germany). Deionized water was also used for cleaning the laboratory tools. The metal stock solutions Cu, Cr, Pb, Se, and Zn were prepared from metal nitrate of 99.99% purity. The glassware was cleaned and rinsed with deionized water.

### 2.2 Preparation of extracts and synthesized Fe-NPs

Banana, pomegranate, orange, potato, onion, and other waste fresh peels were gathered from marketplaces in Giza, Egypt. The production of extracts and the synthesis of Fe-NPs were carried out as seen in Fig 1, according to Niraimathee et al [19]. Characterization of Fe-NPs was assessed using UV-Visible spectroscopy (UV-Vis) (Edinburgh DS5, Scotland) and the Fourier transform infrared (FT-IR) spectral analysis (Spectrum FT-IR Spectrometer; PerkinElmer, Waltham, MA, USA) at wavelengths ranging from 400 to 4000 cm$^{-1}$ as described by Niraimathee et al [19]. The XRD analysis was carried out using a Philips XRD 3100 diffractometer (Amsterdam, Netherlands), as well as x-ray fluorescence (XRF) and energy dispersive x-ray (EDX).

### 2.3 Adsorption isotherm study

Batch experiments were used to measure the adsorption of heavy metal ions at varied concentrations ranging from 1.0 to 10 mg/L. Each test involved filling 100 mL Erlenmeyer flasks with an adsorbent and a solution containing various quantities of heavy metal ions at a solid/solution ratio of 0.4 g/L. The mixture was then agitated for 45 min at 25˚C (Thermo Scientific). Following the reaction time, the concentration of heavy metal ions in each sample was measured after procedure. All experiments were conducted at room temperature, and after being allowed to stand, the samples were analyzed using the atomic absorption spectrometer (AAS) equipped with a graphite furnace to measure very low concentrations (Thermo Fisher Scientific American's provisioned ICE 3000 AAS) [10]. Eqs 1 and 2 were used to determine the amount of metal ions adsorbed at equilibrium (qe) and the percentage removed (R%).

$$R\% = \frac{(Ci - Ce)}{Ci} x\ 100 \tag{1}$$

$$qe = \frac{(Ci - Ce)V}{W} \tag{2}$$

Where $q_e$ (mg/g) represents the amount of ions adsorbed at equilibrium, V (L) is the volume

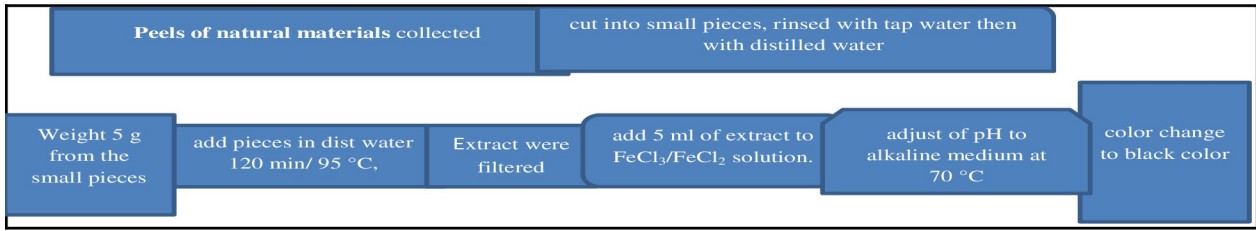

**Fig 1. Schematic diagram of the iron nanoparticles from waste natural waste materials experimental setup.**

of the solution, W (g) is the mass of the nanoparticles used, $C_i$, and $C_e$ (mg/L) represents metal ion concentrations at initial and equilibrium conditions, respectively [20].

**2.3.1. Influence of contact time.** During varied contact times, the adsorption of Cu, Cr, Pb, Se, and Zn onto Fe-NPs was investigated. The studies were conducted at 25˚C or room temperature. The adsorbent dose was 0.3 g, and contact times were 0, 15, 30, 45, and 60 min. The adsorbent was combined with a synthetic aqueous solution at a concentration of 3.0 mg/L.

**2.3.2 Effect of dose on adsorption profile.** The investigation of the effect of adsorbent dose on adsorption capacity is also part of batch studies [21]. The aqueous solution volume utilized in the experiment was 1 L at 150 rpm, the contact period was 45 min, and the initial concentration of heavy metals was 3.0 mg/L. The adsorbent was mixed with an aqueous solution at various dosages (0.1, 0.2, 0.3, 0.4, and 0.5 g) in the combined system.

**2.3.3. Effect of initial concentration.** The experiment was conducted by creating different heavy metal concentrations and holding the temperature constant at 25˚C. It was investigated how the initial concentration of heavy metals affected the effectiveness of adsorption. The adsorbents were mixed with an aqueous solution at concentrations of (1.0, 2.0, 3.0, 5.0, and 10) mg/L with the same 45 min contact time and dose of 0.4 g. The experiment used 1 L of a synthetic aqueous solution rotating at 150 rpm [17].

**2.3.4 Effect of pH.** Among the many variables that affect the metal adsorption process is pH. The pH is the most critical parameter affecting any adsorption studies due to their interference in the solid–solution interface, affecting the charges of the active sites of the adsorbents and the metal behavior in the solution [22]. The effect of pH on the adsorption of Cu, Pb, Se, Zn, and Cr ions in the solution has been established and considered an important parameter affecting the performance of the adsorption process. At varied pH levels (3,6, and 9), removing of Cu, Pb, Se, Zn, and Cr ions from the aqueous solution was conducted with a constant dosage of 0.4 g/L at 150 rpm/45 min [17].

## 2.4 Batch studies using adsorbent

The adsorption isotherms were comprehensively investigated. The effect of operating parameters on the adsorption efficiency such as adsorbent dose, contact time, and initial concentration of metals was examined. The effect of the dose of adsorbent, and concentration of metals on removal efficiency (%), and capacity $q_e$ (mg/g) of metals were characterized [23]. The removal efficiency (R %), the dose of metals adsorbed on a specific dose of adsorbent $q_e$ (mg/g) was calculated from the equations:

$$R\% = \frac{Co - Ce}{Co} \times 100 \tag{3}$$

$$q_e(mg/g) = \frac{(Co - Ce)V}{m} \tag{4}$$

where R % is the removal efficiency, $q_e$ is the dose of metals adsorbed on a specific dose of adsorbent (mg/g), $C_o$ is the initial concentration (mg/L), $C_e$ is the concentration after adsorption (mg/L), m is the dose of adsorbents (g), and V is the volume of solution (Liter).

**2.4.1 Langmuir isotherm model.** Adsorption isotherms, which are typically the ratio between the amount adsorbed and that was left in solution at equilibrium at a specific temperature, are used to describe equilibrium studies that give the capacity of the adsorbent and adsorbate [24]. The Langmuir model is predicated on the hypothesis that maximal adsorption happens in the presence of a saturated monolayer of solute molecules on the adsorbent surface,

the adsorption energy is constant, and there is no adsorbate molecule migration in the surface plane. The Langmuir isotherm model implies that physical factors drive monolayer sorption. The Langmuir isotherm equation is given by:

$$\frac{Ce}{qe} = \frac{1}{qmax.KL} + \frac{Ce}{qmax} \tag{5}$$

$$RL = \frac{1}{1 + Co.KL} \tag{6}$$

where $q_e$ and $K_L$ are the Langmuir constants, where m is the dose of metals adsorbed on a specific dose of adsorbent (mg/g), $C_e$ is the equilibrium concentration of the solution (mg/L) and $q_e$ is the maximum dose of metals concentration required to form a monolayer (mg/g). The values of $q_e$ and $K_L$ can be determined from the linear plot of $C_e/q_e$ versus $C_e$ [25].

**2.4.2 Freundlich isotherm model.**   The Freundlich isotherm model proposes that many sites with various adsorption energies are involved in the empirical relationship that describes the adsorption of solutes from a liquid to a solid surface. The system's characteristics $K_F$ and n are the indicators of the adsorption capacity and adsorption intensity, respectively. The Freundlich model's capacity to match the experimental data was investigated. The intercept value of $K_F$ and the slope of n were calculated for this scenario using the plot of log $C_e$ vs. log $q_e$. The Freundlich isotherms appear when the surface is heterogeneous and the absorption is multilayered and bound to sites on the surface.

$$\log qe = \log KF + \frac{1}{n}\log Ce \tag{7}$$

where $K_F$ is the Freundlich equilibrium constant (mg/g), 1/n = Intensity parameter $C_e$ = Equilibrium concentration of adsorbate, m is the dose of solute adsorbed [26–28]. Freundlich model with linear plotted log qe versus log $C_e$ shown in Eq (6) [10].

The constants $K_F$ and 1/n are produced via the Freundlich formulation in a linear form. Freundlich isotherm model assumes non-ideal adsorption on heterogeneous surfaces in a multilayer coverage. It suggests that more robust binding sites are occupied first, followed by weaker binding sites. In other words, as the degree of site occupation increases, the binding strength decreases.

## 2.5 Kinetic study

It was possible to characterize the kinetics for each adsorbent, using pseudo-first and pseudo-second-order kinetic models. The pseudo-first-order kinetics follows the Lagergren model expressed by Eq 8:

$$\text{Log}\,(q_{eq} - q_t) = \log q_{eq} - K_1 \cdot t/2.303 \tag{8}$$

where qt is the adsorbed dose of metallic ions (mg/g) in t time (min) and k1 is the pseudo-first-order constant (min-1). Through linear and angular constant of log graphic ($q_{eq}$—$q_t$) in the function of time, $q_{eq}$ and $k_1$ can be calculated, respectively. Comparing the experimentally obtained values for $q_{eq}$ calculated by Eq 9:

$$t/q_t = 1/K_2 q_{eq}^{\;2} + /q_{eq} \tag{9}$$

where $k_2$ is the pseudo-second-order constant (g/mg. min) obtained by calculation of linear coefficient and $q_{eq}$ is calculated through angular coefficient.

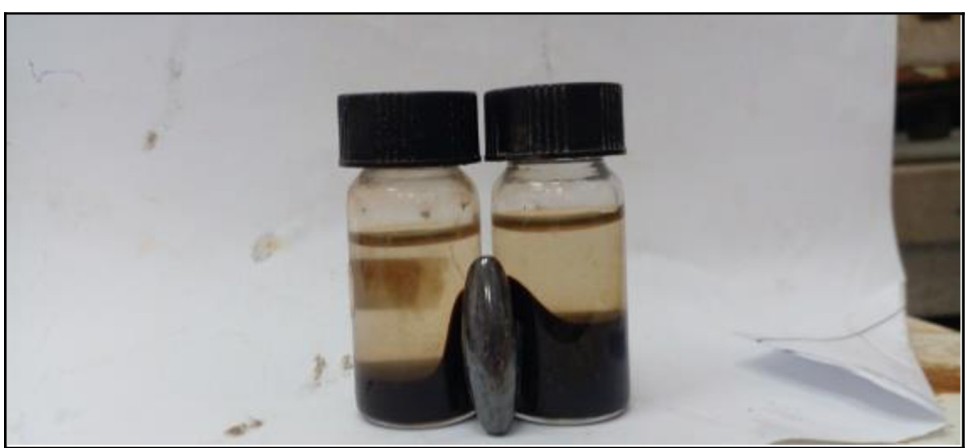

**Fig 2. Extracted Fe-NPs from peels of banana, pomegranate, opuntia, orange, potato, and onion under an external magnetic field.**

## 3. Results and discussion

### 3.1 Characterization of iron nanoparticles

**3.1.1 Super paramagnetism of iron oxide nanoparticles.** The color of the Fe-NP solutions gradually changed to black after 30 min, which clearly indicates the formation of iron oxide colloidal nanoparticles, and the above solution is very clear. In the presence of a magnetic field, the iron oxide nanoparticles demonstrated a magnetic property. Iron oxide nanoparticles were drawn to a magnet when it was positioned close to the glass bottle, as seen in Fig 2. The black nanoparticles were drawn to the magnet, but when the magnetic field was released, shaking was all that was needed to scatter the particles. So, using a straightforward magnetic device, the magnetic nanoparticles can be eliminated or recycled in the medium [19].

**3.1.2 The XRD, and XRF analysis.** The characteristic of banana, orange, opuntia, onion, potato, and pomegranate-Fe-NPs peak occurred around $2\theta = (10°–60°)$. The XRD pattern of the Fe-NP is shown in Fig 3. The structural features of Fe-NPs were explored using XRD data. The peaks found in the XRD pattern were lapelled in banana $2\theta = 33, 35$, and 41, in opuntia, $2\theta = 18, 30, 35$, and 44 in pomegranate $2\theta = 18, 30, 35$, and 43, in orange $2\theta = 30, 35$, and 43, in onion $2\theta = 32$, and 35, in potato $2\theta = 35$. The XRD analysis proved that the iron (III) oxide ($Fe_2O_3$) particles have been successfully aided with extracts of banana, orange, opuntia, onion, potato, and pomegranate. The geometry of the Fe-NPs discovered using the XRD pattern corresponds to $Fe_2O_3$ crystals [29, 30]. The XRF pattern of Fe-NPs prepared from banana, orange, opuntia, onion, potato, and pomegranate is shown in Table 1. The percentage of $Fe_2O_3$ in the Fe-NPs study was 58.9%, 46.6%, 49.0%, 67.3%, 53.9%, and 46.6% for banana, orange, opuntia, onion, potato, and pomegranate, respectively. This indicates that the Fe-NPs of banana, onion, potato, and pomegranate are more efficient than those of other waste materials in this study.

**3.1.3 Fourier transform infrared spectroscopy (FT-IR).** An analysis by infrared spectrophotometer was used to establish the functional groups present in the prepared powders. The nanoparticle powder sample was mixed with potassium bromide (KBr) powder and ground into fine powders. A thin layer of about 0.5 mm, IR cell was used to measure the solution sample. The IR spectra were gathered between 400 and 4000 $cm^{-1}$, as shown in Fig 4. The functional groups were examined and responsible for adsorption process by nanoparticle powder sample, these peaks are 3400 $cm^{-1}$, 2900 $cm^{-1}$, 1600 $cm^{-1}$, 1000 $cm^{-1}$, and 1550 $cm^{-1}$.

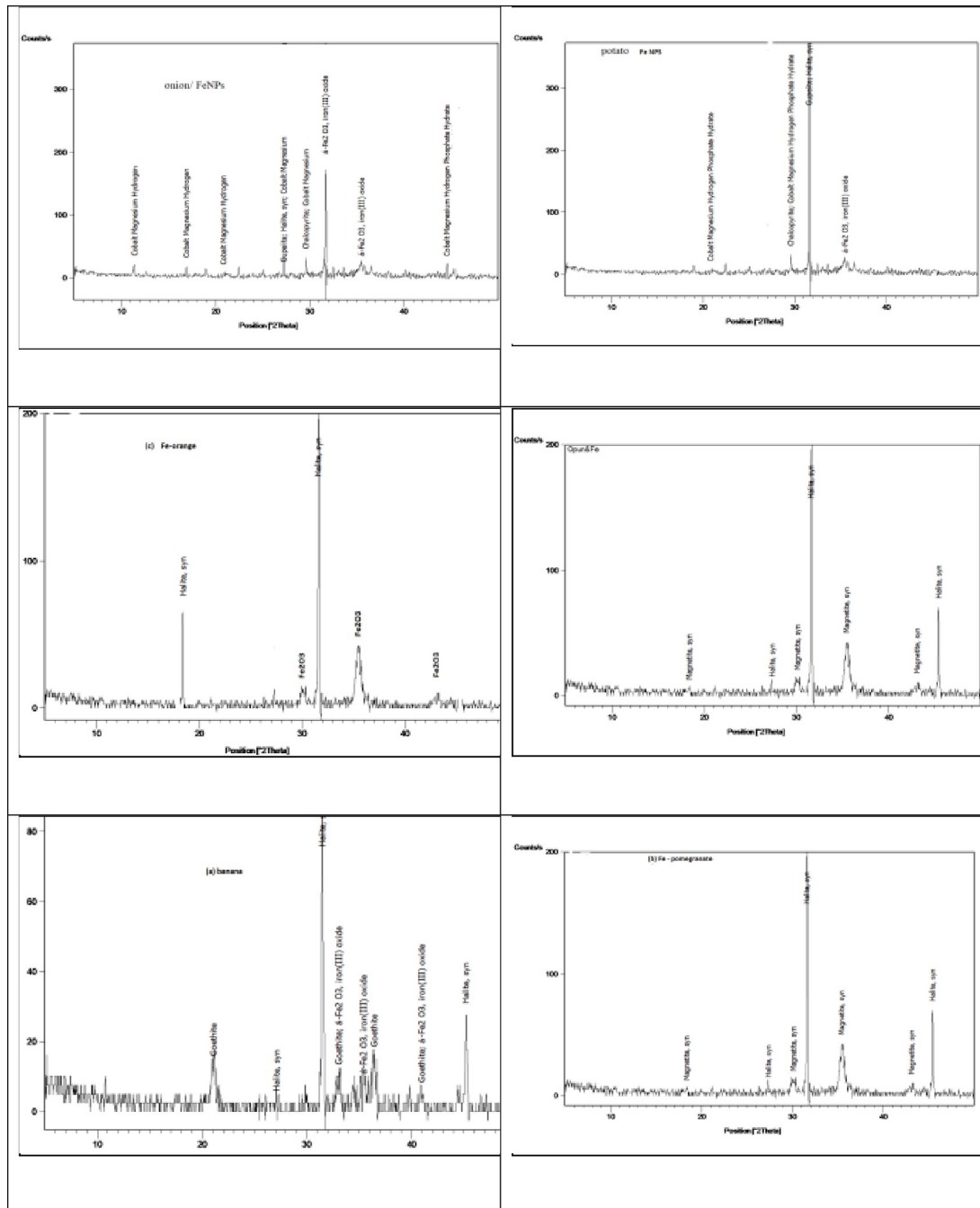

**Fig 3. X- ray diffraction (XRD) pattern of banana, pomegranate, opuntia, orange, potato, and onion extract (%).**

The band at 3400 cm$^{-1}$ in the FT-IR spectrum of banana, pomegranate, opuntia, orange, potato, and onion were related to the O-H bond stretching of the phenolic group. It was suggested that in addition to being responsible for iron reduction, the hydroxyl groups are crucial for iron absorption. When hydroxyl is present in coordinate water molecules, the stretching vibration of functional group OH often creates a large brand region in the range of 3400 cm$^{-1}$ [11]. The methyl-C-H stretching is responsible for the broad peaks at 2900 cm$^{-1}$. The presence

**Table 1. The XRF analysis of Fe-NPs (%).**

| Component | Fe-NPs banana | Fe-NPs Potato | Fe-NPs Onion | Fe-NPs Orange | Fe-NPs Opuntia | Fe-NPs Pomegranate |
|---|---|---|---|---|---|---|
| $Fe_2O_3$ | 58.9 | 53.9 | 67.3 | 46.6 | 49 | 46.6 |
| $Na_2O$ | 14 | 16.1 | 8.39 | 22.1 | 20.1 | 22.1 |
| MnO | 0.46 | 0.53 | 0.63 | 0.46 | 0.46 | 0.46 |
| CaO | 0.28 | 0.3 | 0.47 | 0.27 | 0.27 | 0.27 |
| $SiO_2$ | 0.25 | 0.37 | 0.38 | 0.36 | 0.36 | 0.36 |
| $Al_2O_3$ | 0.1 | 0.09 | 0.08 | 0.07 | 0.07 | 0.07 |
| $K_2O$ | 0.05 | 0.05 | 0.14 | —— | 0.07 | 0.07 |
| $SO_3^-$ | 0.08 | 0.09 | 0.05 | 0.07 | —— | —— |
| $Cr_2O_3$ | 0.05 | 0.05 | 0.05 | —— | 0.04 | 0.04 |
| MgO | 0.04 | 0.04 | 0.04 | 0.04 | ——— | ——— |
| $Cl^-$ | 16.1 | 9.63 | 11.6 | 14.9 | 13.9 | 14.9 |
| LOI | 9.61 | 18.8 | 9.51 | 15.1 | 15.1 | 15.1 |
| Total | 99.94 | 99.98 | 99.94 | 99.99 | 99.37 | 99.99 |

of a new peak at 1600 cm$^{-1}$ is attributed to carbonyl groups (C stretching, vibration = O). The FT-IR spectrum of synthesized Fe-NPs displayed stretching vibrations at 1550 cm$^{-1}$ for C = C, and 1000 cm$^{-1}$ for C–O–C. These adsorption peaks support the presence of protein and other bioactive compounds on the surface of biosynthesized Fe-NPs, confirming that metabolically produced bioactive compounds act as capping agents during production and prevent the reduced iron particles from agglomerating [17]. The appearance of a new frequency peak of >700 cm$^{-1}$ in the spectra of Fe-NPs corresponds to the vibrations of iron oxide's Fe-O bonds [29].

**3.1.4 UV-Vis spectral analysis.** In order to determine if the iron has surface plasmon resonance, the production of nanoparticles was characterized using UV-Vis spectroscopy [17]. The appearance of color arises from the ability of the orange- to black-colored material to absorb selectively within the visible region of the electromagnetic spectrum and scan the spectra between 200 and 600 nm at a resolution of 0.5 nm. The optical properties of iron nanoparticles were determined through UV-Vis region spectra. The optical absorption spectra were recorded. The optical absorption coefficient has been calculated in the wavelength region 190–600 nm. The presence of the maximum absorption band at 386 nm, 386 nm, 400 nm, 420 nm, 210 nm, 215 nm, and 272 nm in the UV-Vis of banana, pomegranate, opuntia, orange, potato, and onion, respectively, as shown in Fig 5. The peaks of the iron oxide nanoparticles were at 386 nm, 386 nm, 400 nm, 420 nm, 210 nm, 215 nm, and 272 nm. The change in the position of the absorption peak of the iron colloidal nanoparticles may be due to the change in the size of the colloidal nanoparticles [18].

**3.1.5 EDX spectrum analyses.** Fig 6 shows the qualitative energy dispersive x-ray (EDX) spectrum studies on the surface of Fe-NPs. The iron (Fe), carbon (C), and oxygen (O) elements' energy patterns for the X-ray character were disclosed by the EDX spectrum. No contaminants were found that could be linked to contamination by chemical precursors. According to some, the heavy metal ions have successfully adhered to the magnetic particles' surface [31].

**3.1.6 Scanning Electron Microscopy (SEM).** Scanning Electron Microscopy (SEM) was used to characterize the surface morphology of adsorbents. The SEM of adsorbents before adsorption is shown in Fig 7. The micrographs, Fig 7, show the porous structures, and the pore sizes of different adsorbents. Because of the available binding cavities for the metal ions, these surface features will result in high metal binding [32].

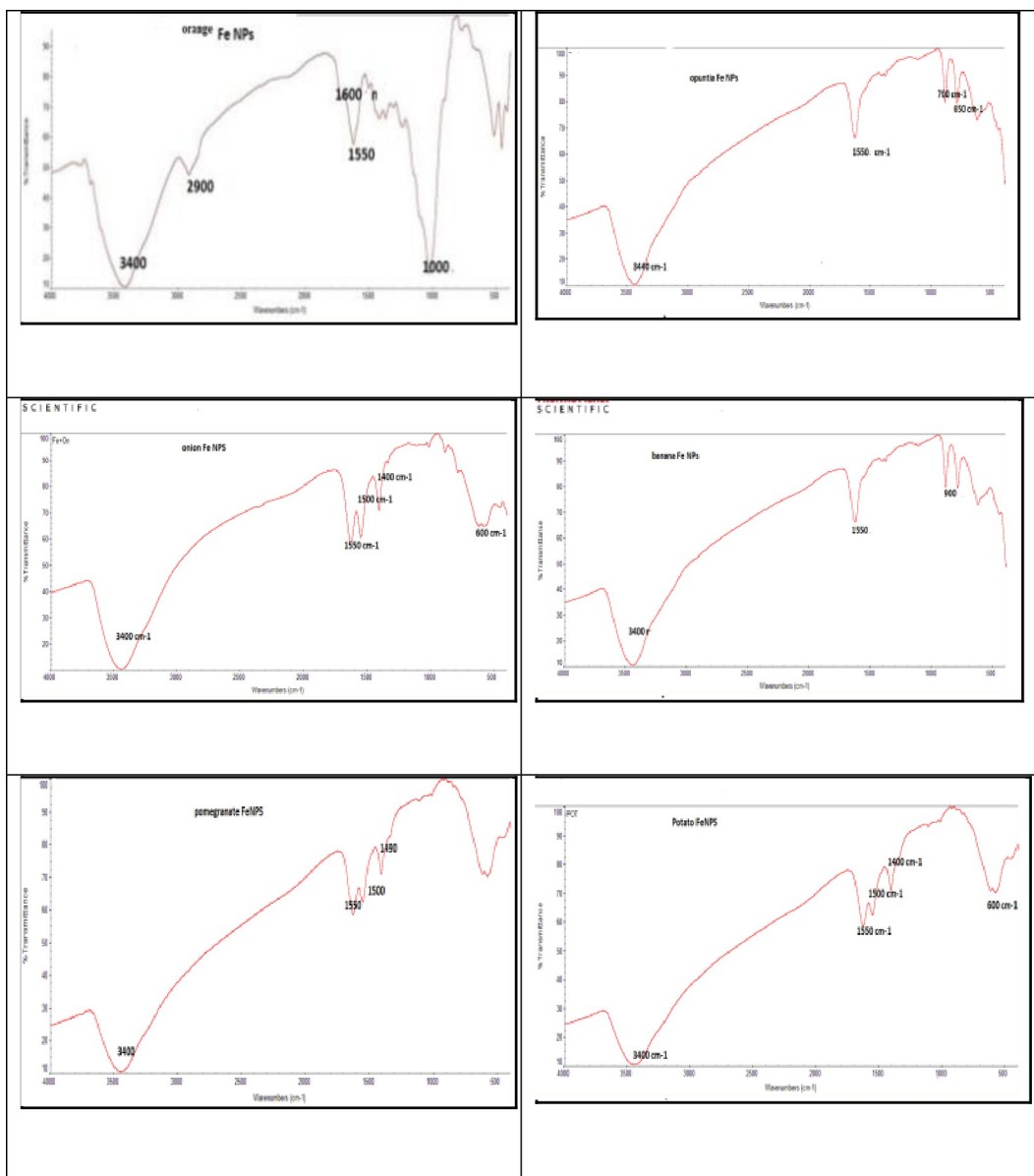

**Fig 4. FTIR spectrums of synthesized Fe-NPs for banana, pomegranate, opuntia, orange, potato, and onion.**

### 3.2 Adsorption study

The adsorption of Cu, Cr, Pb, Se, and Zn onto synthesized Fe-NPs was investigated at different contact times. The studies were conducted at 25˚C or the ambient temperature. The adsorbents were mixed with a 3.0 mg/L synthetic aqueous solution, 0.4 g of adsorbent and different contact times (15, 30, 45, and 60) min at pH 6.0 with a stirring rate of 150 rpm. The removal efficiencies for Cu were 92.86%, 100%, 99.91%, 99.85%, 17.26%, and 88.66% for banana, potato, orange, onion, opuntia, and pomegranate. The results showed that Fe-NPs aided with potato and orange are highly effective adsorbents, with a high removal efficiency of adsorbents at 45 min as shown in Fig 8. The results are in good agreement with that obtained by several studies [33–35].

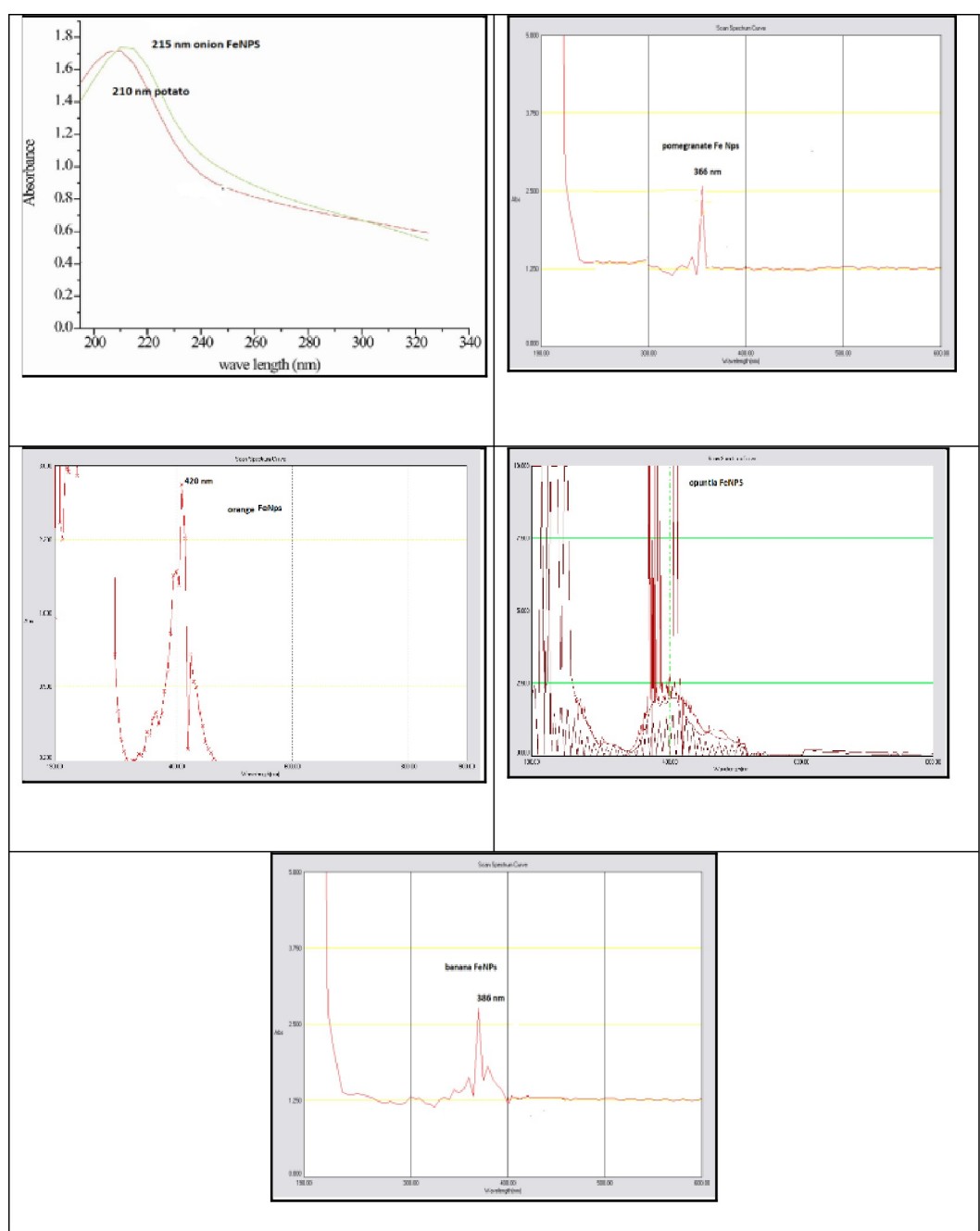

**Fig 5. The UV-Vis for banana, pomegranate, opuntia, orange, potato, and onion.**

The investigation of the effect of adsorbent dose on adsorption capacity was also part of batch studies [21]. In the combined system, the adsorbents were dissolved in an aqueous solution at varied dosages (0.1, 0.2, 0.3, 0.4, and 0.5 g). The initial concentration of heavy metals used in the experiment was 3.0 mg/L, and the aqueous solution volume used was 1 L at 150 rpm. The effect of adsorbent dose on heavy metals removal efficiency was investigated for Cu, Cr, Pb, Se, and Zn concentrations of 3.0 mg/L at (0.1, 0.2, 0.3, 0.4, and 0.5 g) Fe-NPs dosage at pH 3.0 for 45 min with stirring rate of 150 rpm at room temperature 25°C. The removal

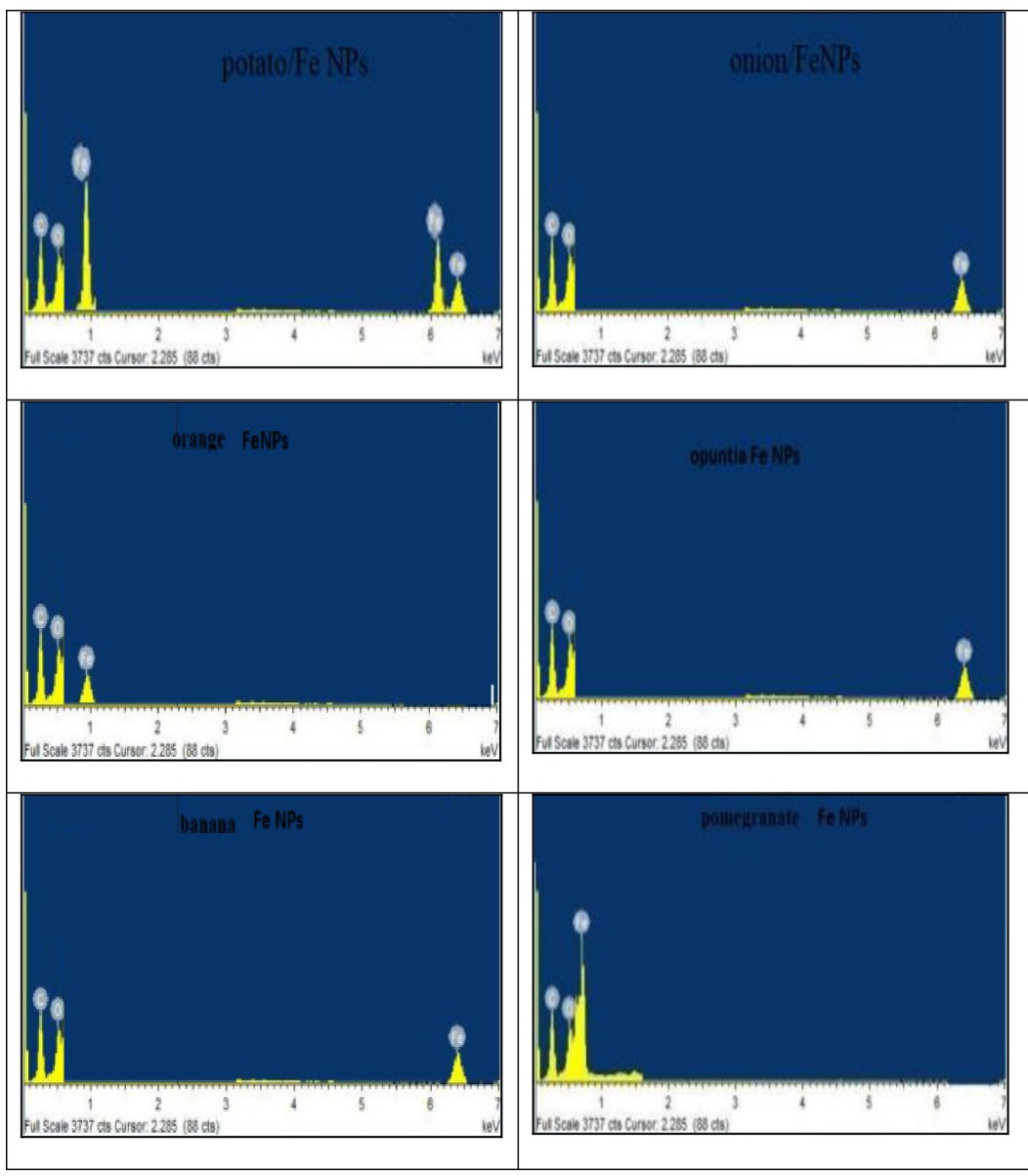

**Fig 6. The EDX spectra of the magnetic adsorbent for banana, pomegranate, opuntia, orange, potato, and onion.**

efficiencies for Cu were 99.94%, 100%, 100%, 100%, 20.00%, and 90.00% for banana, potato, orange, onion, opuntia, and pomegranate, respectively. While, the removal efficiencies for Cr were 91.46%, 100%, 92.90%, 93.40%, 88.86%, and 100% for banana, potato, orange, onion, opuntia, and pomegranate, respectively; removal efficiencies for Pb were 99.95%, 100%, 100% as shown in Fig 9. The results showed that the effective dose of Fe-NPs was 0.4 g. The removal efficiency increased with dose due to an increase in the vacant site for adsorption and free electrons for the degradation process. Different studies conducted with metal ion removal using different adsorbent materials with different doses showed high-efficiency properties for the reduction of heavy metal concentrations under different operating conditions [36, 37].

By preparing various concentrations of the heavy metals at 25°C, it was possible to assess the impact of an initial concentration of heavy metals on the adsorption effectiveness of

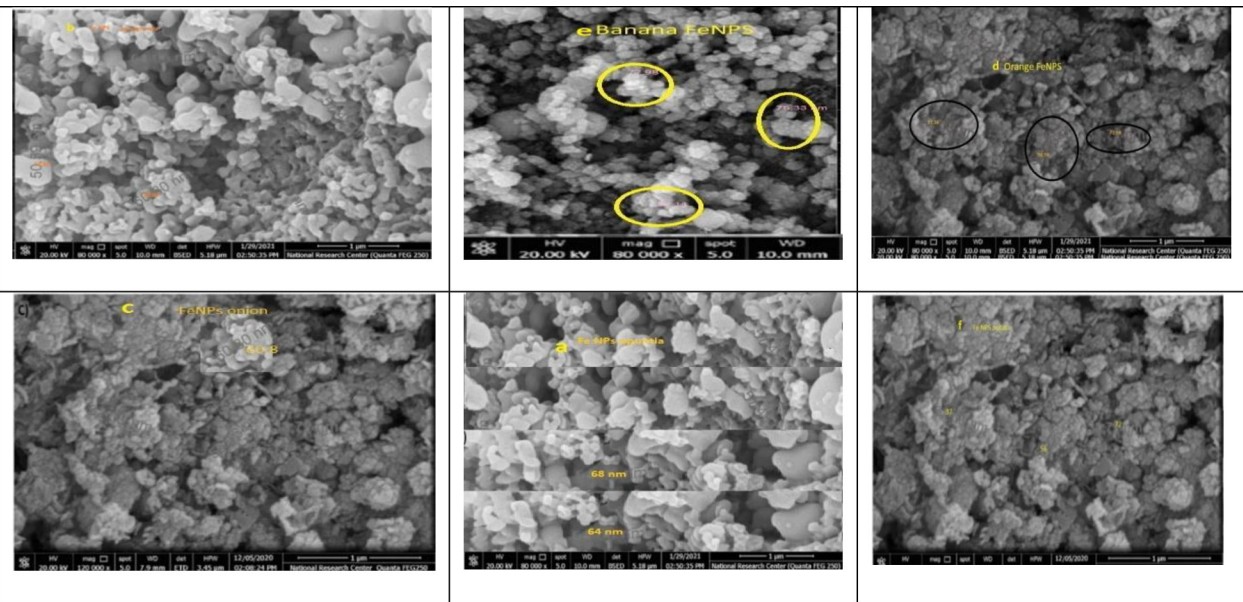

**Fig 7.** The SEM images of the surface iron nanoparticles of (a) opuntia and (b) pomegranate (c) onion, (d) orange,) e) banana, and (f) potato extract.

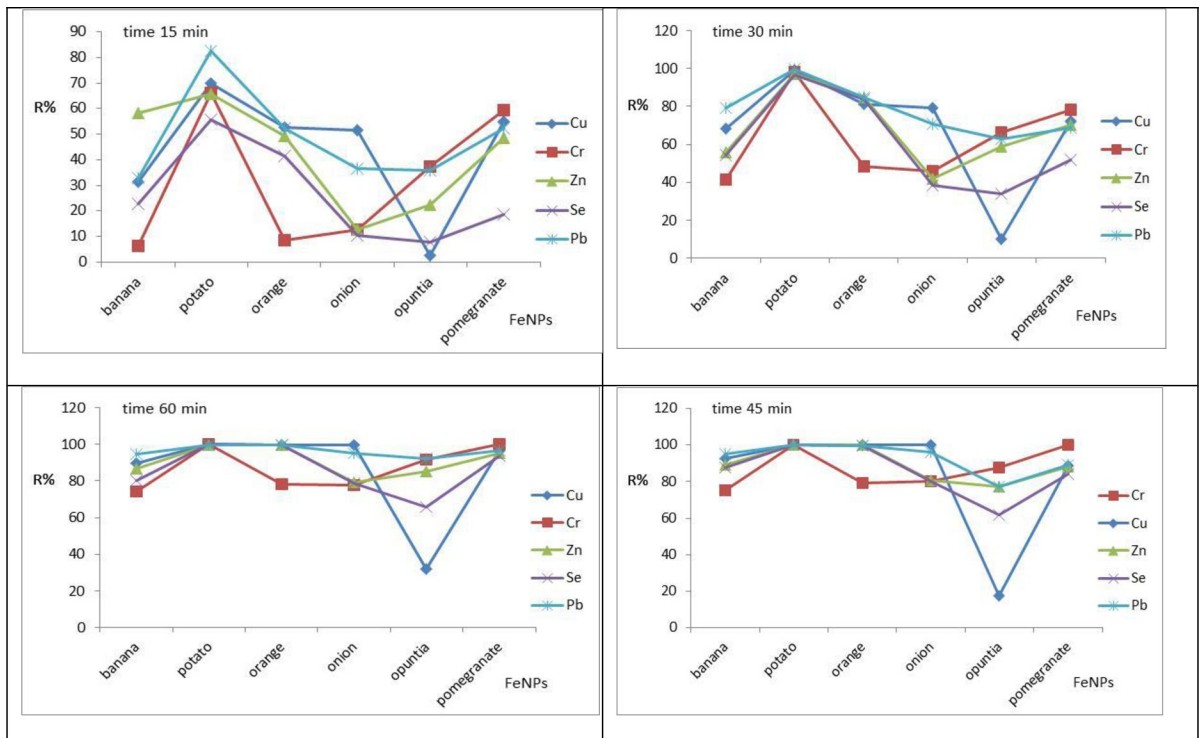

**Fig 8. Influence of contact time on Cu, Cr, Pb, Se, and Zn removal using Fe NPs: At Ph, 3.0, adsorbents initial concentration 3.0 mg/l, adsorbents doses 0.3 g, stirring rate 150 rpm.**

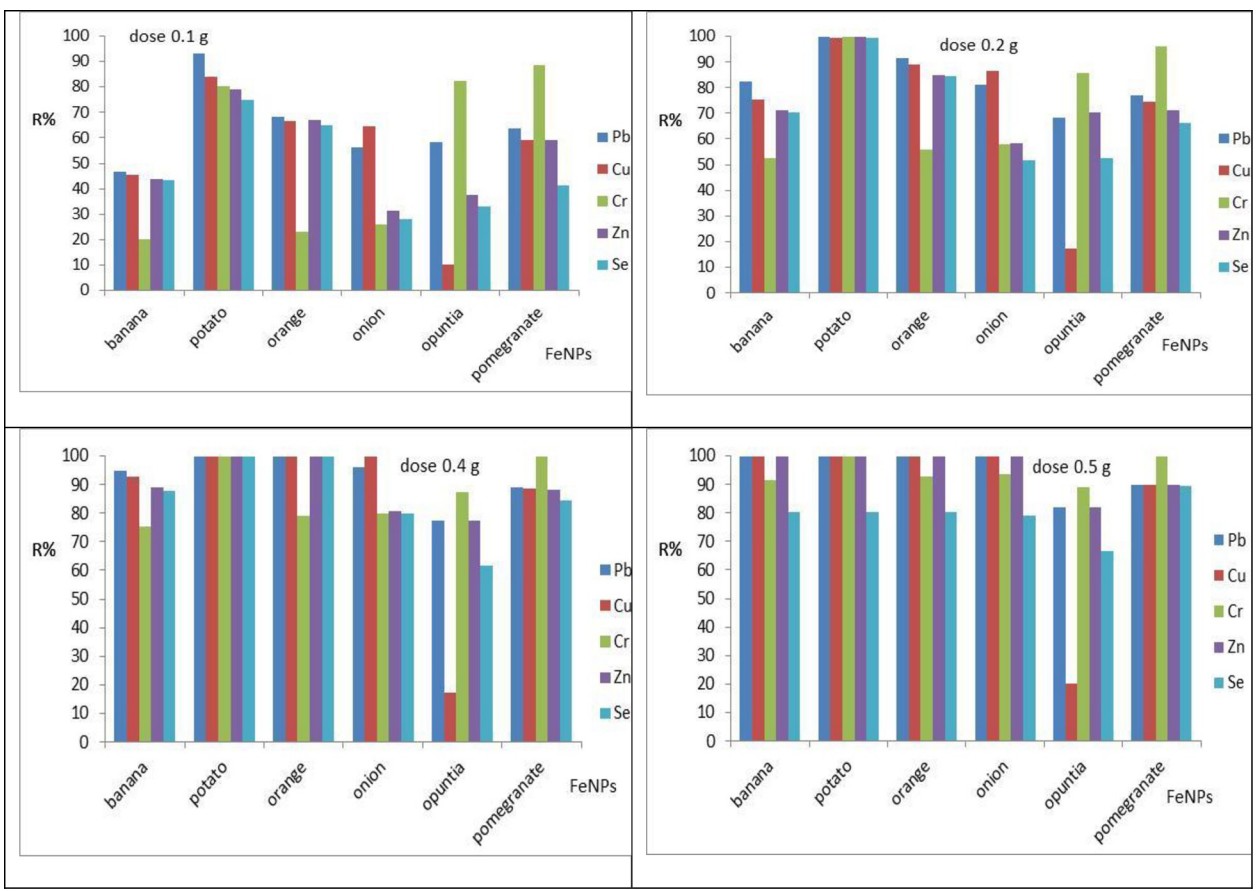

**Fig 9. Effect of adsorbents doses on Cu, Cr, Pb, Se, and Zn removal using Fe NPs: At pH, 3.0, adsorbents initial concentration 3.0 mg/l, stirring rate 150 rpm, and contact time 45 min.**

adsorbents. At concentrations of (0.5, 1.0, 2.0, 3.0, 4.0, and 5.0 mg/L), at contact times of 45 min, and at doses of 0.4 g/l, the adsorbents were combined with aqueous solutions. 1 L of a synthetic aqueous solution rotating at 150 rpm was utilized in the experiment [17, 34, 35, 38].

The effect of initial concentration on heavy metals removal efficiency was investigated for Cu, Cr, Pb, Se, and Zn concentrations (1.0, 2.0, 3.0, 5.0, and 10.0) mg/L using Fe-NPs dosage of 0.4 g/l at pH 3.0 for 45 min with stirring rate 150 rpm. The removal efficiencies at 1.0 mg/L for Cu were 100%, 100%, 100%, 56.00%, and 96.00% for banana, potato, orange, onion, opuntia, and pomegranate, respectively. The removal efficiency for Pb were 100%, 100%, 100%, 100%, 90.00%, and 96.00% for banana, potato, orange, onion, opuntia, and pomegranate, respectively. The removal efficiency for Zn was 99.85%, 100%, 100%, 98.46%, 88.00%, and 96.50% for banana, potato, orange, onion, opuntia, and pomegranate, respectively. The removal efficiencies for Se were 95.80%, 100%, 100%, 96.80%, 81.00%, and 90.20% for banana, potato, orange, onion, opuntia, and pomegranate, respectively. Fig 10 shows the influence of the initial concentration of metal ions. The removal efficiency opuntia is low due to in the vacant site for adsorption and free electrons for the degradation process is very low other than adsorbents.

The maximum adsorption capacity obtained from the equilibrium studies was in the following order: Cu (6.96, 7.50, 7.49, 7.48, 1.29, and 6.65) mg/g, Pb (7.12, 7.5, 7.48, 7.19, 5.80, and 6.67) mg/g, Zn (6.69, 7.50, 7.48, 6.06, 5.80, and 6.62) mg/g, Cr (5.63, 7.50, 5.93, 6.00, 6.55, and

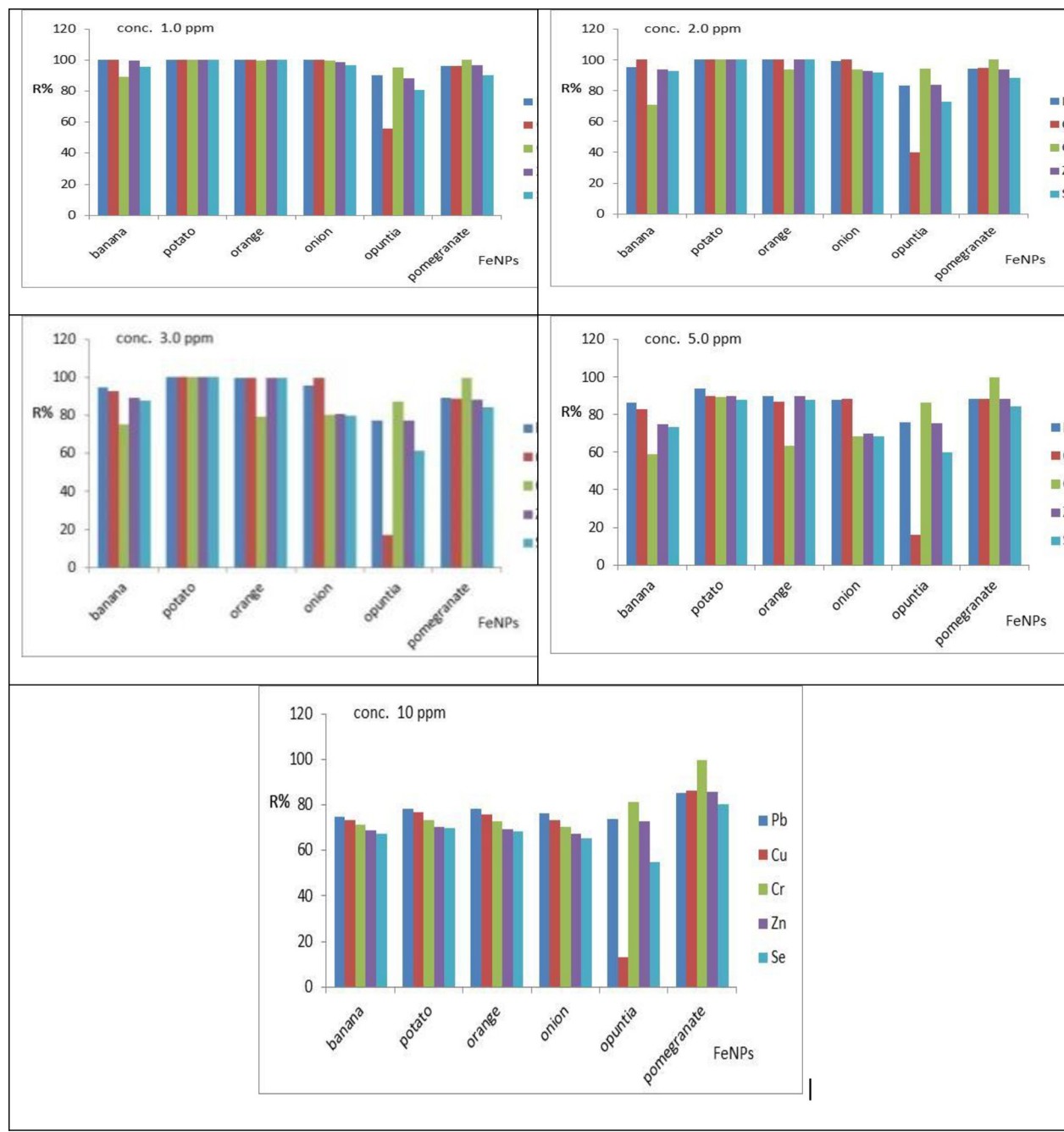

**Fig 10. Performance of Fe-NPs for removal of Cu, Cr, Pb, Se, and Zn as affected by the initial concentrations at fixed operating conditions (pH, 3.0, adsorbents doses 0.4 g, contact time 45 min, stirring rate 150 rpm).**

7.49) mg/g, and Se (6.58, 7.50, 7.48, 5.99, 4.61, and 6.63) mg/g for banana, potato, orange, onion, opuntia, and pomegranate, respectively. The discrepancy in radius and interaction enthalpy values may be the cause of the various sorption capabilities. Table 2 compares the Fe-NPs adsorbent's maximal adsorption capacity with various adsorbents described in the literature for adsorption. The equilibrium concentrations of the metal ions in the solution were shown to cause the ferromagnetic sorbent's adsorption capacity to rise, gradually approaching

**Table 2. The Fe-NPs adsorbent's maximal adsorption capacity is compared to that of other adsorbents used in the removal of metal ions.**

| Adsorbents | Adsorbents | Q max | Reference |
|---|---|---|---|
| Untreated rice husk | Direct dyes | 2.4 | [40] |
| Activated rice husk | Direct dyes | 4.3 | [40] |
| Red-mud | $Ni^{+2}$ | 0.0018 | [41] |
| Peanut Hulls | $Fe^{+3}$ and $Cu^{+2}$ | 79.28 and 96.58 mg/g for $Fe^{+3}$ and $Cu^+$ | [42] |
| Zeolite derived from fly ash | $Cu^{+2}$ | 14.7 | [43] |
| Ag nanoparticle-loaded activated carbon (Ag-NP-AC) | $Cu^{+2}$ | 60 | [44] |
| Iron oxide coated eggshell powder | $Cu^{+2}$ | 6.7 | [45] |
| Chitosan | $Cu^{+2}$ | 62.4 | [46] |

saturation. The findings showed that as the concentration of metal ions in the solution increased, so did the concentration difference between the bulk solution and the surface, accelerating the mass transfer processes [33]. This sorption feature shows that the initial metal ion concentrations-controlled surface saturation. Adsorption sites quickly absorbed the available metal at low concentrations, but at high concentrations, metal ions had to diffuse to the sorbent surface via intra-particle diffusion, and highly hydrolyzed ions diffused more slowly. The metal ions were first diffused onto the sorbent surface during the adsorption process from the boundary layer film, and then they were diffused into the sorbent's porous structure [39].

**3.2.1 The Influence of pH on the adsorption study.** The pH dependence of metal ions' uptake was linked to both the surface functional groups and the metal ion species predominant in an aqueous solution. The species metals (M) and M(OH) are predominant at pH lower than 6, while the groups on the surface are protonated and cannot bind to metal ions in the solution.

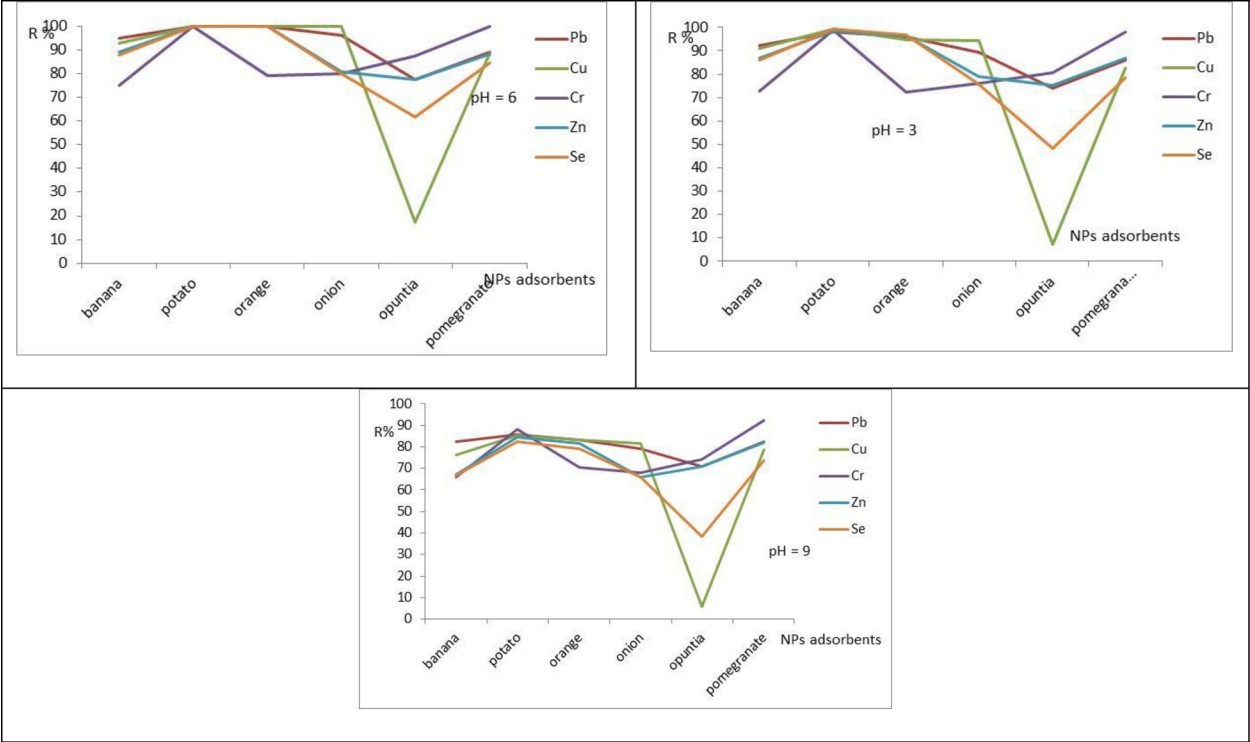

**Fig 11. Effect of pH on removal of Cu, Pb, Se, Zn, and Cr by iron nanoparticles of banana, potato, orange, onion, opuntia, and pomegranate, the other conditions kept constant, the during the adsorption processes dose = 04 g/L, Initial conc. = 3.0 mg/L and rpm 150/45 min.**

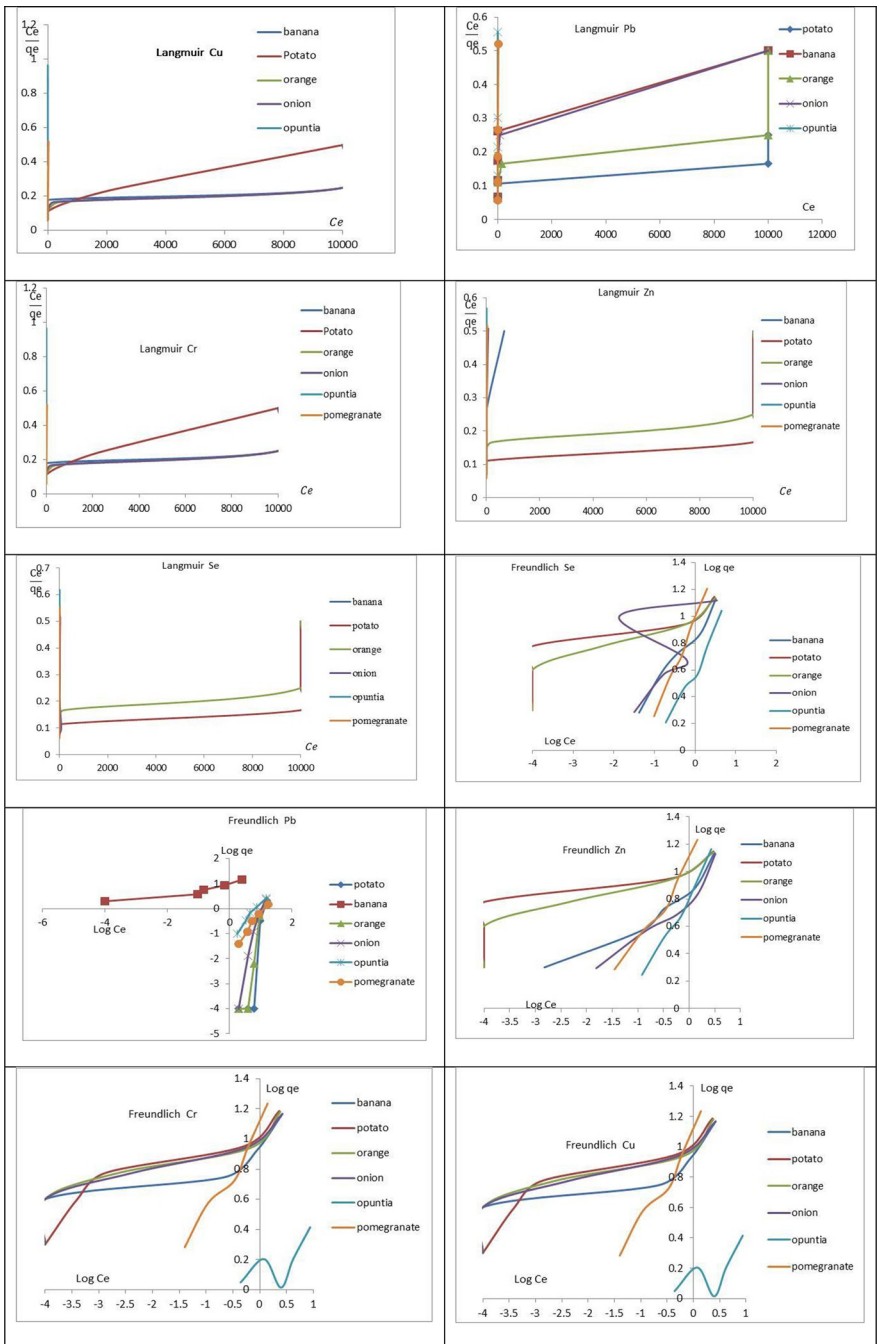

**Fig 12. The Langmuir and Freundlich isotherms models of Pb; Cr Se Cu, and Zn onto iron nanoparticles of banana, potato, orange, onion, opuntia, and pomegranate, the other conditions kept constant, the during the adsorption processes dose = 0.4 g/L, and rpm 150/45 min.**

Besides, at very low pH, the surface groups are associated with the hydronium ions ($H_3O^+$), negatively affecting the interaction with the metal cations. When the pH increases, the surface affinity with the metal also increases, and adsorption is improved [47]. One of the essential factors for the sequestration of heavy metal ions by adsorption from aqueous solutions is the initial solution pH because it influences the charge on the adsorbent surface, the degree of

**Table 3. The parameters of Langmuir and Freundlich isotherm models for Cu, Se, Zn, Pb, and Cr adsorption on the banana, potato, orange, onion, opuntia, and pomegranate.**

| Langmuir isotherm models | $R_2$ | | | | | $q_{max}$ | | | | | $K_L$ | | | | |
|---|---|---|---|---|---|---|---|---|---|---|---|---|---|---|---|
| | Pb | Cu | Cr | Zn | Se | Pb | Cu | Cr | Zn | Se | Pb | Cu | Cr | Zn | Se |
| Fe NPs (Banana) | 0.7345 | 0.7767 | 0.7954 | 0.8237 | 0.9674 | 25000 | 33333 | 33333 | 23 | 22 | 0.0002 | 0.66 | 0.76 | 0.73 | 0.75 |
| Fe NPs (Potato) | 0.4878 | 0.9621 | 0.9621 | 0.4749 | 0.4709 | 25000 | 33653 | 33245 | 49595 | 50000 | 0.0003 | 0.0002 | 0.0002 | 0.0002 | 0.0002 |
| Fe NPs (Orange) | 0.6942 | 0.6439 | 0.6436 | 0.6955 | 0.6936 | 33333 | 33457 | 34873 | 34332 | 33333 | 0.0002 | 0.0002 | 0.0002 | 0.0002 | 0.0002 |
| Fe NPs (Onion) | 0.8393 | 0.6949 | 0.6949 | 0.8667 | 0.0017 | 25000 | 34233 | 33333 | 21 | 20 | 0.0002 | 0.0014 | 0.13 | 0.16 | 0.81 |
| Fe NPs (Opuntia) | 0.3581 | 0.2771 | 0.2778 | 0.9704 | 0.9478 | 20000 | 44 | 43 | 20 | 20 | 0.0006 | 0.0005 | 0.13 | 0.98 | 1.0 |
| Fe NPs (Pomegranate) | 0.964 | 0.9661 | 0.966 | 0.941 | 0.9869 | 58 | 55 | 58 | 21 | 22 | 0.18 | 0.16 | 0.18 | 0.64 | 0.62 |
| Freundlich isotherm models | $R^2$ | | | | | N | | | | | $K_F$ | | | | |
| | Pb | Cu | Cr | Zn | Se | Pb | Cu | Cr | Zn | Se | Pb | Cu | Cr | Zn | Se |
| Fe NPs (Banana) | 0.8554 | 0.9791 | 0.9792 | 0.9056 | 0.9789 | 5.53 | 5.33 | 6.33 | 7.01 | 7.22 | 9.0 | 9.40 | 9.13 | 8.7 | 6.5 |
| Fe NPs (Potato) | 0.889 | 0.977 | 0.9873 | 0.7134 | 0.7042 | 0.24 | 0.34 | 0.42 | 0.98 | 1.55 | 343 | 12.3 | 66 | 54 | 44 |
| Fe NPs (Orange) | 0.876 | 0.8702 | 0.8679 | 0.8701 | 0.8661 | 0.21 | 0.33 | 0.54 | 6.98 | 6.87 | 345 | 12.3 | 22 | 14 | 13 |
| Fe NPs (Onion) | 0.9449 | 0.8722 | 0.8679 | 0.9332 | 0.1545 | 0.20 | 6.8 | 7.40 | 7.35 | 7.44 | 388 | 12.3 | 55 | 14 | 12 |
| Fe NPs (Opuntia) | 0.9757 | 0.9772 | 0.4986 | 0.988 | 0.9767 | 0.64 | 4.4 | 4.46 | 3.65 | 3.98 | 55 | 1.3 | 1.3 | 1.5 | 1.3 |
| Fe NPs (Pomegranate) | 0.9831 | 0.9775 | 0.9725 | 0.9726 | 0.9961 | 0.59 | 1.7 | 1.72 | 1.56 | 1.66 | 66 | 13 | 11 | 1.4 | 11 |

ionization, and the species of adsorbates [48]. In the current research work, the influence of pH on the removal of Cu, Pb, Se, Zn, and Cr ions from aqueous solution at constant initial concentration was carried out at pH of 3, 6, and 9. The percentage removal of Cu, Pb, Se, Zn, and Cr ions was significantly higher at pH 6 than at lower pH values for all adsorbents as shown in Fig 11. Higher adsorption at pH 6.0 may be attributed to the presence of a larger number of vacant sites for biosorption of Cu, Pb, Se, Zn, and Cr ions in the acidic medium. Additionally, the number of negatively charged functional groups that were present on the surface of the bio-sorbent for binding of Cu, Pb, Se, Zn, and Cr ions increased at optimum pH. This resulted in a decreased struggle between protons and metal ions [8, 48, 49]. The percentage of Cu, Pb, Se, Zn, and Cr ion removal was low at pH 3.0, and 9.0 during the adsorption process because the solution was acidic [7, 8], highly alkaline, but at pH 6.0 the removal efficiency was very high [2, 7]. The maximum removal efficiency of Cu, Pb, Se, Zn, and Cr ion for the case of adsorbents were 82.2%, 93%, 81.55%, 82%, and 82% at pH = 6.0 for banana, potato, orange, onion, opuntia, and pomegranate, respectively [50, 51].

## 3.3 Adsorption isotherms

The adsorption isotherms of the studied metals on the iron nanoparticles for extracts of banana, potato, orange, onion, opuntia, and pomegranate, were based on the optimum operating conditions which were 0.4 g at pH 6.0 Fig 12. The linearization was performed according to the mathematical models of Freundlich. Table 3 shows the parameters of Langmuir and Freundlich models obtained and the correlation coefficients of adsorption data. The experimental results of Cu, Pb, Cr, Zn, and Se adsorption on the banana, potato, orange, onion, opuntia, and pomegranate comply with the Freundlich isotherm model according to $R^2$ studied [52]. The model that best fitted for metal adsorption was Freundlich, indicating that the adsorption occurred in multiple layers [52].

The calculated parameters for the Langmuir and Freundlich models are shown in Table 3, along with the correlation coefficients for the adsorption data. According to the experimental results of Cu, Pb, Cr, Zn, and Se adsorption on the iron nanoparticles of banana, potato, orange, onion, opuntia, and pomegranate, the adsorption took place in multiple layers and

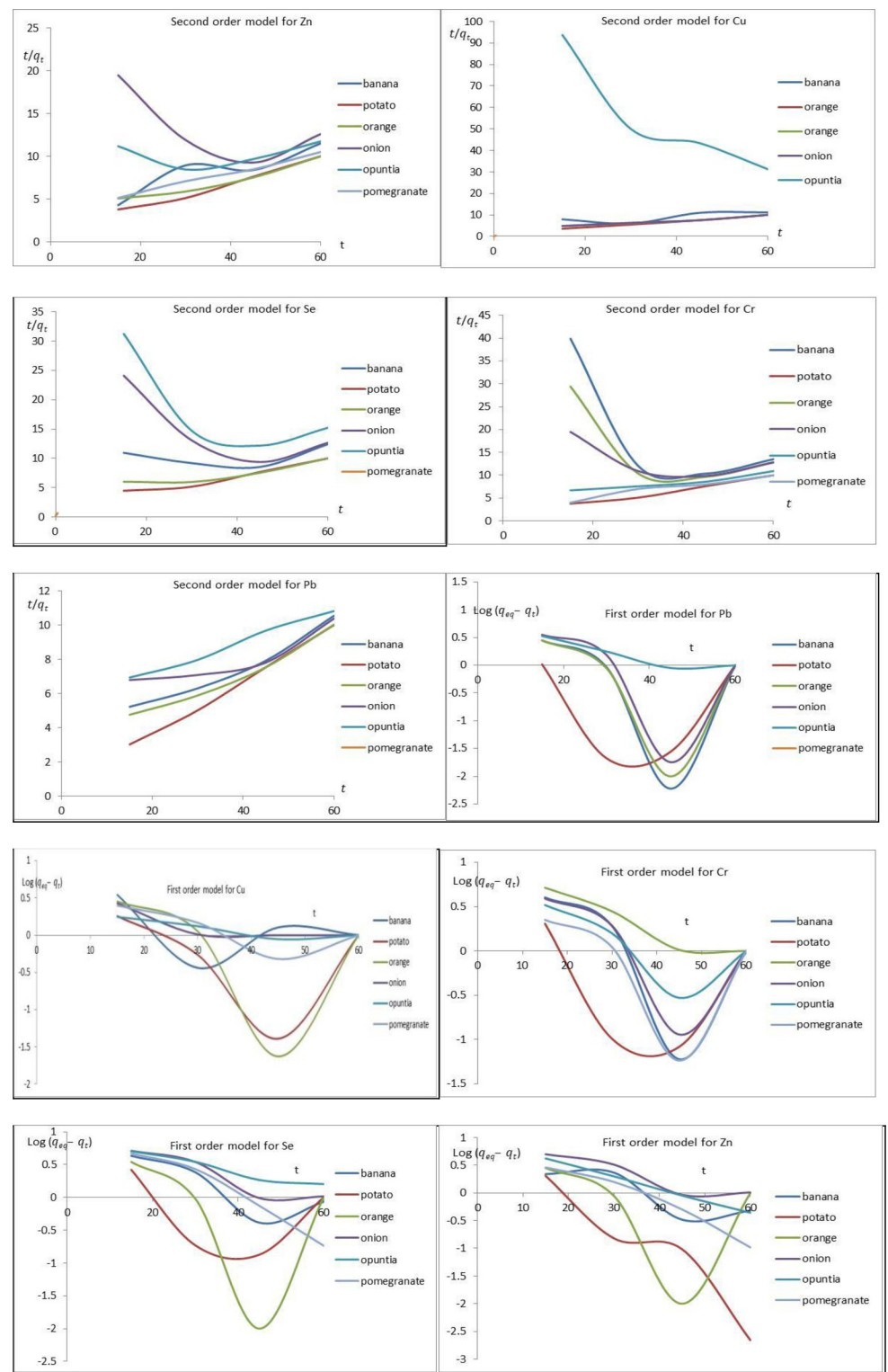

**Fig 13. The pseudo-first-order and pseudo-second-order of Pb, Cr, Se, Cu, and Zn onto iron nanoparticles of banana, potato, orange, onion, opuntia, and pomegranate, the other conditions kept constant, the during the adsorption processes dose = 04 g/L, initial concentration 3.0 mg/L, and speed 150 rpm.**

**Table 4. The parameters of pseudo-second-order model parameters of Cu, Se, Zn, Pb, and Cr adsorption onto iron nanoparticles of banana, potato, orange, onion, opuntia, and pomegranate.**

| Adsorbents/Metal | $R^2$ | | | | | Q | | | | | $K_2$ | | | | |
|---|---|---|---|---|---|---|---|---|---|---|---|---|---|---|---|
| | Pb | Cu | Cr | Zn | Se | Pb | Cu | Cr | Zn | Se | Pb | Cu | Cr | Zn | Se |
| FeNPs/Potato | 0.9974 | 0.9974 | 0.9974 | 0.9134 | 0.9001 | 10.8 | 10.75 | 7.14 | 7.14 | 7.78 | 0.001 | 0.001 | 0.015 | 0.01 | 0.01 |
| FeNPs/Banana | 0.9425 | 0.9712 | 0.9567 | 0.9223 | 0.9113 | 10.43 | 10.30 | 1.88 | 1.88 | 1.98 | 0.001 | 0.001 | 0.007 | 0.007 | 0.007 |
| FeNPs/Onion | 0.9325 | 0.9711 | 0.225 | 0.9012 | 0.913 | 9.11 | 9.01 | 8.33 | 8.33 | 8.76 | 0.003 | 0.004 | 0.0008 | 0.0008 | 0.0007 |
| FeNPs/Opuntia | 0.9911 | 0.9468 | 0.9911 | 0.915 | 0.9563 | 0.87 | 0.83 | 11.11 | 100 | 101 | 0.011 | 0.013 | 0.0016 | 0.00001 | 0.00001 |
| FeNPs/Orang | 0.9971 | 0.9717 | 0.9442 | 0.9334 | 0.9553 | 11.28 | 11.23 | 3.03 | 3.03 | 3.12 | 0.001 | 0.001 | 0.003 | 0.003 | 0.004 |
| FeNPs/pomegrante | 0.9974 | 0.9525 | 0.9627 | 0.9011 | 0.991 | 11.87 | 11.36 | 8.33 | 8.33 | 8.34 | 0.001 | 0.0015 | 0.005 | 0.003 | 0.003 |

was consistent with the Freundlich isotherm model [52]. The metals Cu, Cr, Se, and Zn have higher adsorption capacities ($q_m$) and the highest binding energies with the adsorbent in the Langmuir linearization. As the adsorbents ranged from 1.3 to 388 (mg/g), the values of $K_F$ for Cu, Se, Zn, Pb, and Cr were calculated from Table 3. The characteristics of each metal and the manner of interaction with the adsorbents can be linked to this adsorption sequence. The magnitudes of $K_F$ demonstrate the simple removal of metal ions from the aqueous solution and suggest an advantageous adsorption process [52–54].

## 3.4. Kinetic study

Fig 13 shows the kinetics models for Cu, Pb, Se, Zn, and Cr ion adsorption onto banana, potato, orange, onion, opuntia, and pomegranate. The relation between $\log (q_{eq}\text{-}q_t)$ and t. The $K_1$ and $q_e$ are obtained from the slope and intercept, respectively. The correlation coefficients ($R^2$) of the pseudo-first-order kinetic model are non-linear but the pseudo-second-order kinetic model linear line, the correlation coefficients ($R^2$) of the pseudo-second-order is shown in Table 4. The results suggested that the overall rates of adsorption of Cu, Pb, Se, Zn, and Cr ions onto banana, potato, orange, onion, opuntia, and pomegranate were controlled by chemisorption [55]. Besides, the value of $R^2$ (linear correlation) around 1 in pseudo-second-order confirms that the adsorption kinetics is controlled by this order and that there is a strong interaction between adsorbent and adsorbate.

Fig 13 shows the kinetics models for Cu, Pb, Se, Zn, and Cr ion adsorption onto banana, potato, orange, onion, opuntia, and pomegranate. The pseudo-first-order kinetic model shows

**Table 5. Material and energy consumption for production of 0.1 kg of adsorbents.**

| Process | Water consumption* ($m^3$) | Electricity consumption* (kWh) |
|---|---|---|
| **Washing** | 0.002 | - |
| **Drying at $105^0$ C (24h)** | - | 2 |
| **Crushing and sieving** | - | 0.5 |
| **FeCl$_3$ 1 kg** | | |
| **FeCl$_2$ 1 kg** | | |
| **Waste materials 1 kg** | | |
| **Washing** | 0.0015 | - |
| **Total Consumption** | 0.0035 | 2.5 |
| **Cost** | 0.035 | 3.5 |

*In Egypt, the cost of 1 $m^3$ of water for industrial use = 10.0 L.E

*In Egypt, the cost of 1 kWh of electricity for industrial use = 1.45 L.E

in Fig 13, that's shows the relation between log ($q_{eq}$-$q_t$) and t, and through this relation the values $K_1$ and $q_e$ could be deduced from the slope and intercept, respectively. The correlation coefficients ($R^2$) of the pseudo-second-order kinetic model are higher than the pseudo-first-order kinetic model, and $q_e$ values calculated from the pseudo-second-order kinetic model are very close to the experimental with Freundlich isotherm model. The results suggested that the overall rates of adsorption of Cu, Pb, Se, Zn, and Cr ion onto adsorption onto banana, potato, orange, onion, opuntia, and pomegranate by physical adsorption [55].

### 3.5 Cost of adsorbents

Based upon the preparation process used in this study, the cost analysis of using banana, potato, orange, onion, opuntia, and pomegranate as effective adsorbents of heavy metals from an aqueous solution was calculated as shown in Table 5. The cost analysis shows that the specific energy consumption of the adsorbent production is 5.5 kWh/$m^3$ and water consumption is 0.045 $m^3$. The cost needed for the production of 0.14 kg of adsorbent is 3.935 Egyptian Pound, which is equivalent to \$0.137. This is considered a very low value as adsorbents produced from an nano materials at a priceless cost than other adsorbents such as activated carbon or other adsorbents [56]. Besides being a cost-effective treatment method for hazardous pollutants such as heavy metals, the adsorption process is environmentally friendly and does not generate secondary byproducts. The cost was calculated according to the following: The cost for 100 g of adsorbents = cost of materials + cost of electricity consumed. Hence, the cost = 0.045 + 3.935 = 3.980 L.E. for the production of 100g.

## 4. Conclusion

The results of the current study showed that the green synthesis-produced Fe-NPs adsorbent was effective in removing the heavy metal ions (Cu, Zn, Pb, Se, and Cr) from aqueous solutions. The preparation of iron nanoparticles were characterized many distinct magnetic properties, the ability to control their size, morphology, and high surface area properties, have many function group responsible for adsorption process.

The batch studies revealed that the adsorption process was determined by contact time, dose, and initial concentration at stirring rates of 150 rpm with a solution pH of 3.0. For preparing iron nanoparticles, the green synthesis was simple and eco-friendly. The nanoparticles formed from extracting natural waste materials aided by $FeCl_2$, and $FeCl_3$ aqueous solutions have controlled sizes and crystal structures depending on the type of waste materials used, such as banana, pomegranate, opuntia, orange, potato, and onion. The XRD analysis demonstrates the nanoparticle had the $Fe_2O_3$ crystal form. The superparamagnetic Fe-NPs were successfully synthesized from the peel extracts of banana, pomegranate, opuntia, orange, potato, and onion. The superparamagnetic property of the Fe-NPs at room temperature was confirmed by the magnetic measurements. The Fe-NPs were synthesized by reduction of an iron solution using banana, pomegranate, opuntia, orange, potato, and onion peel extracts, which act as the reducing agent. The involvement of functional groups presents in the biomolecules responsible for the reduction of iron oxide nanoparticles was revealed by the FT-IR spectrum. The functional groups were examined and responsible for adsorption process by nanoparticle powder sample, these peaks are (3400 $cm^{-1}$, 2900 $cm^{-1}$, 1600 $cm^{-1}$, 1000 $cm^{-1}$, and 1550 $cm^{-1}$).

## Author Contributions

**Data curation:** Nowarah J. Almutlq.

**Formal analysis:** Mutairah S. Al Shammari, Hussein M. Ahmed, Fatehy M. Abdel-Haleem, Nowarah J. Almutlq, Mohamed A. El-Khateeb.

**Funding acquisition:** Mutairah S. Al Shammari.

**Investigation:** Mohamed A. El-Khateeb.

**Methodology:** Mohamed A. El-Khateeb.

**Writing – original draft:** Mohamed A. El-Khateeb.

**Writing – review & editing:** Mohamed A. El-Khateeb.

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
