## [Decision Letter · Decision Letter 0]

29 Mar 2023

PONE-D-23-05787Adsorption of Chromium, Copper, Lead, Selenium, and Zinc Ions into Ecofriendly Synthesized Magnetic Iron NanoparticlesPLOS ONE

Dear Dr. El-Khateeb,

Thank you for submitting your manuscript to PLOS ONE. After careful consideration, we feel that it has merit but does not fully meet PLOS ONE’s publication criteria as it currently stands. Therefore, we invite you to submit a revised version of the manuscript that addresses the points raised during the review process.

We look forward to receiving your revised manuscript.

Kind regards,

Vusumzi Pakade

Academic Editor

PLOS ONE

Journal Requirements:

Review of Green Methods of Iron Nanoparticles Synthesis and Applications - https://doi.org/10.1007/s12668-018-0516-5

Synthesis and Characterization of Iron Oxide Nanoparticle by Precipitation Method - https://www.arcjournals.org/pdfs/ijarps/v2-i8/6.pdf

(among others)

In your revision ensure you cite all your sources (including your own works), and quote or rephrase any duplicated text outside the methods section. Further consideration is dependent on these concerns being addressed.

   "The authors extend their appreciation to the Deputyship for Research & Innovation, Ministry of Education in Saudi Arabia for funding this research work through project number 223202." 

   "The authors extend their appreciation to the Deputyship for Research & Innovation, Ministry of Education in Saudi Arabia for funding this research work through project number 223202"

   "The authors extend their appreciation to the Deputyship for Research & Innovation, Ministry of Education in Saudi Arabia for funding this research work through project number 223202."

6. Please ensure that you refer to Figure 1 in your text as, if accepted, production will need this reference to link the reader to the figure.

Additional Editor Comments:

Academic Editor's comments:

1) Provide a conclusion in the abstract

2) pH is one of the most critical parameters influencing the adsorption of contaminants as it affects the surface chemistry and ionization of species among other things. In the current paper, various metal ions were studies and these may have different optimum pH for the removal by the nanoparticles composites. I strongly suggest that authors include a pH study in their manuscript. 

3) Also, it is important that experiments are reproducible. How many replicates were used by the authors?

4) SEM and TEM analysis must be included in the revised manuscript.

5) Section 2.2 must reference Figure 1.

6) Section 2.3, describe the filter used. Is there any chance that it might have participated in the adsorption? Also considering that low concentrations of pollutants were investigated?

7) Standardize the units throughout the manuscript.

8) Sections 2.3.1, 2.3.2 and 2.3.3 can all be combined into one section. 

9) Section 3.1.2. Delete the repeating sentence

10) The mechanism of the adsorption process must be explained/

11) Kinetic models and adsorption isotherms must be used to interpret the data.

Reviewers' comments:

Reviewer's Responses to Questions

**Comments to the Author**

1. Is the manuscript technically sound, and do the data support the conclusions?

Reviewer #1: No

2. Has the statistical analysis been performed appropriately and rigorously? 

Reviewer #1: No

3. Have the authors made all data underlying the findings in their manuscript fully available?

Reviewer #1: No

4. Is the manuscript presented in an intelligible fashion and written in standard English?

Reviewer #1: Yes

5. Review Comments to the Author

Reviewer #1: The manuscript “Adsorption of Chromium, Copper, Lead, Selenium, and Zinc Ions into Ecofriendly Synthesized Magnetic Iron Nanoparticles” does not meet the minimum required scientific standard for publication in PLOS ONE. Although the work is very specific, the synthesis of magnetic Iron nanoparticles using eco-friendly methods is well studied in previous publications. The following major comments can be clarified to improve acceptability. See the comments below:

1. Section 1. The author suggests “The particle size, magnetic properties, and morphology of Fe-NPs depend on the conditions of the material” but fails to prove the statement in the characterization. SEM and /or TEM are strongly recommended for morphology and particle size measurement.

2. Section 2.2 “The production of extracts and the synthesis of Fe-NPs were carried out according to Niraimathee et al [19]”. Include the method and modification done.

3. Section 2.3 – it is not clear whether the adsorption of heavy metals was simultaneous or individual.

4. Section 3.1.2 - The inclusion of SEM results will show how morphology changes with different modifications.

5. Section 3.1.4 - “The change in the position of the absorption peak of the iron colloidal nanoparticles may be due to the change in the size of the colloidal nanoparticles [18]”. TEM results will support the changes in particle size.

6. Section 3.1.5 – The author should include EDX spectra after adsorption.

7. The author should include TGA results to assess the thermal stability of adsorbents.

8. Section 3.2.1 and 3.2.2 – Explain a significant drop in copper adsorption in figure 7 (17.26 & 20.00% respectively).

9. Can the authors compare the adsorption capacity of the adsorbents with other reported similar adsorbents?

10. The author should include pH and temperature studies as heavy metals behave differently in solution.

11. The authors should give some more information on the mechanism of interaction of the heavy metals with adsorbents, which could be in terms of the ionic forms present and the surface of the powder since they are pH dependent. The author should include the isotherm model, kinetics and thermodynamics to understand the adsorption mechanism.

12. Are these results reproducible with different batches of the adsorbents? Have you done the error analysis? Is there any effect on the size of the powder?

13. A regeneration study is important to evaluate the cost-effectiveness of the adsorbents.

6. PLOS authors have the option to publish the peer review history of their article (what does this mean?). If published, this will include your full peer review and any attached files.

Reviewer #1: No

<quillbot-extension-portal></quillbot-extension-portal>

---

## [Author Response · Author response to Decision Letter 0]

9 May 2023

Dear Valued Reviewers

Thank you for your valuable comments.

All the comments were responded within the text with highlights. Also a separate file is submitted. 

PONE-D-23-05787

Adsorption of Chromium, Copper, Lead, Selenium, and Zinc Ions into Ecofriendly Synthesized Magnetic Iron Nanoparticles

Academic Editor's comments:

Provide a conclusion in the abstract (Done)

2) pH is one of the most critical parameters influencing the adsorption of contaminants as it affects the surface chemistry and ionization of species among other things. In the current paper, various metal ions were studies and these may have different optimum pH for the removal by the nanoparticles Done

2.3.4 Effect of pH

3) Also, it is important that experiments are reproducible. How many replicates were used by the authors? (Three time) Done

4) SEM and TEM analysis must be included in the 3.1.6 Scanning Electron Microscopy (SEM) and TEM not applicable

5) Section 2.2 must reference Figure 1. Done

6) Section 2.3, describe the filter used. Is there any chance that it might have participated in the adsorption? Also considering that low concentrations of pollutants were investigated? Done

7) Standardize the units throughout the manuscript. Done

8) Sections 2.3.1, 2.3.2 and 2.3.3 can all be combined into one section. Done

9) Section 3.1.2. Delete the repeating sentence Done

10) The mechanism of the adsorption process must be explained/ isotherm, kinetic Done, 2.4 Batch studies using adsorbent

11) Kinetic models and adsorption isotherms must be used to interpret the data. Done 

 2.5 Kinetic Study 

Reviewer #1:

1. Section 1. The author suggests “The particle size, magnetic properties, and morphology of Fe-NPs depend on the conditions of the material” but fails to prove the statement in the characterization. SEM and /or TEM are strongly recommended for morphology and particle size measurement. 3.1.6 Scanning Electron Microscopy (SEM) and TEM not applicable

2. Section 2.2 “The production of extracts and the synthesis of Fe-NPs were carried out according to Niraimathee et al [19]”. Include the method and modification done.

3. Section 2.3 – it is not clear whether the adsorption of heavy metals was simultaneous or individual. All metals dissolved in same solution (simultaneous)

4. Section 3.1.2 - The inclusion of SEM results will show how morphology changes with different modifications. Done 

 3.1.6 Scanning Electron microscope

5. Section 3.1.4 - “The change in the position of the absorption peak of the iron colloidal nanoparticles may be due to the change in the size of the colloidal nanoparticles [18]”. TEM results will support the changes in particle size. TEM, not available

6. Section 3.1.5 – The author should if include EDX spectra after adsorption. The EDX for materials FeNPs before done but after absorption Not, available

7. The author should include TGA results to assess the thermal stability of adsorbents. TGA,Not,available

8. Section 3.2.1 and 3.2.2 – Explain a significant drop in copper adsorption in figure 7 (17.26 & 20.00%respectively) The removal efficiency opuntia is low due to in the vacant site for adsorption and free electrons for the degradation process is very low other than adsorbents. 

9. Can the authors compare the adsorption capacity of the adsorbents with other reported similar adsorbents? Table 6: Adsorption efficiencies of some selected adsorbents

10. The author should include pH and temperature studies as heavy metals behave differently in solution 2.3.4 Effect of pH, and temperature not applicable

11. The authors should give some more information on the mechanism of interaction of the heavy metals with adsorbents, which could be in terms of the ionic forms present and the surface of the powder since they are pH dependent. The author should include the isotherm model, kinetics and thermodynamics to understand the adsorption mechanism. 2.5 Kinetic Study 

2.4 Batch studies using adsorbent

2.3.4 Effect of pH

12. Are these results reproducible with different batches of the adsorbents? Done, Three time

Have you done the error analysis? Is there any effect on the size of the powder? Not applicable

13. A regeneration study is important to evaluate the cost-effectiveness of the adsorbents Done 

3.6 Cost of adsorbents

Best Regards

---

## [Decision Letter · Decision Letter 1]

30 May 2023

PONE-D-23-05787R1Adsorption of Chromium, Copper, Lead, Selenium, and Zinc Ions into Ecofriendly Synthesized Magnetic Iron NanoparticlesPLOS ONE

Dear Dr. El-Khateeb,

Thank you for submitting your manuscript to PLOS ONE. After careful consideration, we feel that it has merit but does not fully meet PLOS ONE’s publication criteria as it currently stands. Therefore, we invite you to submit a revised version of the manuscript that addresses the points raised during the review process.

We look forward to receiving your revised manuscript.

Kind regards,

Vusumzi Pakade

Academic Editor

PLOS ONE

Journal Requirements:

Reviewers' comments:

Reviewer's Responses to Questions

**Comments to the Author**

1. If the authors have adequately addressed your comments raised in a previous round of review and you feel that this manuscript is now acceptable for publication, you may indicate that here to bypass the “Comments to the Author” section, enter your conflict of interest statement in the “Confidential to Editor” section, and submit your "Accept" recommendation.

Reviewer #1: All comments have been addressed

Reviewer #2: (No Response)

2. Is the manuscript technically sound, and do the data support the conclusions?

Reviewer #1: Yes

Reviewer #2: Partly

3. Has the statistical analysis been performed appropriately and rigorously? 

Reviewer #1: Yes

Reviewer #2: No

4. Have the authors made all data underlying the findings in their manuscript fully available?

Reviewer #1: Yes

Reviewer #2: Yes

5. Is the manuscript presented in an intelligible fashion and written in standard English?

Reviewer #1: Yes

Reviewer #2: Yes

6. Review Comments to the Author

Reviewer #1: The authors have addressed major concerns raised and the manuscript is recommended for publication in PLOS.

Reviewer #2: This manuscript evaluates the ability of synthesized sorbents (Magnetic Iron Nanoparticles, Fe-NPs) derived from the peel extracts of banana, pomegranate, opuntia, orange, potato, and onion for efficient removal of Pb, Se, Cu, Zn, and Cr from aqueous solutions. The authors claimed that the prepared Fe-NPs adsorbents were effective in removing the studied heavy metals. However, the following technical aspects of the present work are lacking and were note appropriately addressed:

7. PLOS authors have the option to publish the peer review history of their article (what does this mean?). If published, this will include your full peer review and any attached files.

Reviewer #1: No

Reviewer #2: No

While revising your submission, please upload your figure files to the Preflight Analysis and Conversion Engine (PACE) digital diagnostic tool, https://pacev2.apexcovantage.com/. PACE helps ensure that figures meet PLOS requirements. To use PACE, you must first register as a user. Registration is free. Then, login and navigate to the UPLOAD tab, where you will find detailed instructions on how to use the tool. If you encounter any issues or have any questions when using PACE, please email PLOS at figures@plos.org. Please note that Supporting Information files do not need this step.<quillbot-extension-portal></quillbot-extension-portal>

---

## [Author Response · Author response to Decision Letter 1]

24 Jun 2023

There are file attached reveals the respond to the comments of the reviewers.

---

## [Editor Report · Decision Letter 2]

27 Jun 2023

PONE-D-23-05787R2Adsorption of Chromium, Copper, Lead, Selenium, and Zinc Ions into Ecofriendly Synthesized Magnetic Iron NanoparticlesPLOS ONE

Dear Dr. El-Khateeb,

Thank you for submitting your manuscript to PLOS ONE. After careful consideration, we feel that it has merit but does not fully meet PLOS ONE’s publication criteria as it currently stands. Therefore, we invite you to submit a revised version of the manuscript that addresses the points raised during the review process.

I was tempted to accept this manuscript as the reviewers have suggested but upon inspection I noted a number of details that need to be addressed. Hence, I am sending it back to major revision. I advise that authors pay due diligence to the comments made and address them as best as they could. Also, I suggest that authors find a critical reader to assist them with the manuscript packaging.Here are my concerns:1) The comments about quantitative data and conclusion in the abstract were not addressed.2) The Ce in the abstract is 1.0 mg/L, in the conclusion is 3.0 mg/L and in the authors response narrative is 6.0 mg/L. So which is which?3) Section 2.4, what is qmax?4) Section 2.4.1, Equation 3 is incorrect.5) The KL and KF values of the Langmuir and Freundlich must be properly addressed, not loosely as K.6) Table 1 title, that is XRF data not XRD.7) Table 2, the data shown there is incorrect.8) Table 3 and 4, why only show R2 values? Why not other parameters are not shown? Consider putting other data in the supporting document.9) Section 3.5 does not have data and it should be omitted and reported as such that reusability studies were not done. Also, in the authors comments, it is suggested repeatability was done 5.72 times. How is this possible? Fraction?10) Table 6 should compare the adsorption capacity of the current study to literature not just listing the literature values alone.11) What significant FTIR peaks were observed? This can be used to suggest successful synthesis and/ or adsorption. This value (s) can be quoted in the conclusion and abstract.12) Again, what properties of Fe-NPs were explored or came to the fore during adsorption? This was asked before by a reviewer and needs to be included in the conclusion.13) Authors are advised to scrutinize their manuscript even for comments not made here and make necessary adjustments. 

We look forward to receiving your revised manuscript.

Kind regards,

Vusumzi Pakade

Academic Editor

PLOS ONE

Additional Editor Comments:

All comments are provided in the email above.

<quillbot-extension-portal></quillbot-extension-portal>

---

## [Author Response · Author response to Decision Letter 2]

3 Jul 2023

Dear Valued Editor,

Thank you for your help and support. Your comments are helpful and improving the article a lot. The following Table shows the response to these comments.

1 The comments about quantitative data and conclusion in the abstract were not addressed.

2 The Ce in the abstract is 1.0 mg/L, in the conclusion is 3.0 mg/L and in the authors response narrative is 6.0 mg/L. So which is which? 3.0 mg/L

modified

3 Section 2.4, what is qmax qe (mg/g) modified

4 Section 2.4.1, Equation 3 is incorrect Done - delete

5 The KL and KF values of the Langmuir and Freundlich must be properly addressed, not loosely as K. Done 

6 Table 1 title, that is XRF data not XRD Done 

Table 1: The XRF analysis of Fe-NPs (%)

7 Table 2, the data shown there is incorrect. Done modified 

Table 2: The Fe-NPs adsorbent's maximal adsorption capacity is compared to that of other adsorbents used in the removal of metal ions

8 Table 3 and 4, why only show R2 values? Why not other parameters are not shown? Consider putting other data in the supporting document.

 Done 

Table 3: The parameters of Langmuir and Freundlich isotherm models for Cu, Se, Zn, Pb, and Cr adsorption on the banana, potato, orange, onion, opuntia, and pomegranate

Table 4: The parameters of pseudo-second-order model parameters of Cu, Se, Zn, Pb, and Cr adsorption onto iron nanoparticles of banana, potato, orange, onion, opuntia, and pomegranate

9 Section 3.5 does not have data and it should be omitted and reported as such that reusability studies were not done. Also, in the authors comments, it is suggested repeatability was done 5.72 times. How is this possible? Fraction?

 Done – delete

10 Table 6 should compare the adsorption capacity of the current study to literature not just listing the literature values alone. Table 6 was deleted and corporated into Table 2

11 What significant FTIR peaks were observed? This can be used to suggest successful synthesis and/ or adsorption. This value (s) can be quoted in the conclusion and abstract. Done 

The functional groups were examined and responsible for adsorption process by nanoparticle powder sample, these peaks are 3400 cm−1, 2900 cm-1, 1600 cm−1,1000 cm−1, and 1550 cm−1. 

12 Again, what properties of Fe-NPs were explored or came to the fore during adsorption? This was asked before by a reviewer and needs to be included in the conclusion. Done 

The preparation of iron nanoparticles was characterized many distinct magnetic properties, the ability to control their size, morphology, and high surface area properties, have many functions group responsible for adsorption process

13 13) Authors are advised to scrutinize their manuscript even for comments not made here and make necessary adjustments. Done 

Table 1: The XRF analysis of Fe-NPs (%)

Component Fe-NPs

banana Fe-NPs Potato Fe-NPs Onion Fe-NPs Orange Fe-NPs

Opuntia Fe-NPs

Pomegranate 

Fe2O3 58.9 53.9 67.3 46.6 49 46.6

Na2O 14 16.1 8.39 22.1 20.1 22.1

MnO 0.46 0.53 0.63 0.46 0.46 0.46

CaO 0.28 0.3 0.47 0.27 0.27 0.27

SiO2 0.25 0.37 0.38 0.36 0.36 0.36

Al2O3 0.1 0.09 0.08 0.07 0.07 0.07

K2O 0.05 0.05 0.14 ---- 0.07 0.07

SO3-- 0.08 0.09 0.05 0.07 ---- ----

Cr2O3 0.05 0.05 0.05 ---- 0.04 0.04

MgO 0.04 0.04 0.04 0.04 ----- -----

Cl- 16.1 9.63 11.6 14.9 13.9 14.9

LOI 9.61 18.8 9.51 15.1 15.1 15.1

Total 99.94 99.98 99.94 99.99 99.37 99.99

Table 2: The Fe-NPs adsorbent's maximal adsorption capacity is compared to that of other adsorbents used in the removal of metal ions

Adsorbents Adsorbants Q max Reference

Untreated rice husk Direct dyes 2.4 (Abdelwahab, El Nemr et al. 2005)

Activated rice husk Direct dyes 4.3 (Abdelwahab, El Nemr et al. 2005)

Red-mud Ni+2 0.0018 (Lakshmi Narayan, Govindan et al. 2013)

Peanut Hulls Fe+3 and Cu+2 79.28 and 96.58 mg/g for Fe+3 and Cu+ (Salam, Reiad et al. 2011)

Zeolite derived from fly ash Cu+2 14.7 (Mishra and Tiwari 2006) 

Ag nanoparticle-loaded activated carbon (Ag-NP-AC) Cu+2 60 (Shahamirifard, Ghaedi et al. 2016)

Iron oxide coated eggshell powder Cu+2 6.7 (Liu, Liang et al. 2005)

Chitosan Cu+2 62.4 (Crini 2005)

 

Figure 11: Effect of pH on removal of Cu, Pb, Se, Zn, and Cr by iron nanoparticles of banana, potato, orange, onion, opuntia, and pomegranate, the other conditions kept constant, the during the adsorption processes dose=04 g/L, Initial conc.= 3.0 mg/L and rpm 150/45 min

3.3 Adsorption Isotherms 

The adsorption isotherms of the studied metals on the iron nanoparticles for extracts of banana, potato, orange, onion, opuntia, and pomegranate, were based on the optimum operating conditions which were 0.4 g at pH 6.0 Figure 12. The linearization was performed according to the mathematical models of Freundlich. Table 3 shows the parameters of Langmuir and Freundlich models obtained and the correlation coefficients of adsorption data. The experimental results of Cu, Pb, Cr, Zn, and Se adsorption on the banana, potato, orange, onion, opuntia, and pomegranate comply with the Freundlich isotherm model according to R2 studied (Gonçalves Junior, Meneghel et al. 2013). The model that best fitted for metal adsorption was Freundlich, indicating that the adsorption occurred in multiple layers(Gonçalves Junior, Meneghel et al. 2013).

The calculated parameters for the Langmuir and Freundlich models are shown in Table 3, along with the correlation coefficients for the adsorption data. According to the experimental results of Cu, Pb, Cr, Zn, and Se adsorption on the iron nanoparticles of banana, potato, orange, onion, opuntia, and pomegranate, the adsorption took place in multiple layers and was consistent with the Freundlich isotherm model (Gonçalves Junior, Meneghel et al. 2013). The metals Cu, Cr, Se, and Zn have higher adsorption capacities (qm) and the highest binding energies with the adsorbent in the Langmuir linearization. As the adsorbents ranged from 1.3 to 388 (mg/g), the values of KF for Cu, Se, Zn, Pb, and Cr were calculated from Table 3. The characteristics of each metal and the manner of interaction with the adsorbents can be linked to this adsorption sequence. The magnitudes of KF demonstrate the simple removal of metal ions from the aqueous solution and suggest an advantageous adsorption process (Kumar and Kirthika 2009, Gonçalves Junior, Meneghel et al. 2013, dos Santos, de Toledo Gomes et al. 2019).

Figure 12: The Langmuir and Freundlich isotherms models of Pb; Cr Se Cu, and Zn onto iron nanoparticles of banana, potato, orange, onion, opuntia, and pomegranate, the other conditions kept constant, the during the adsorption processes dose=04 g/L, and rpm 150/45 min

Table 3: The Langmuir and Freundlich isotherm models for Cu, Se, Zn, Pb, and Cr adsorption on the banana, potato, orange, onion, opuntia, and pomegranate

Langmuir isotherm models R2 qmax KL

 Pb Cu Cr Zn Se Pb Cu Cr Zn Se Pb Cu Cr

 Zn Se

Fe NPs (Banana) 0.7345 0.7767 0.7954 0.8237 0.9674 25000 33333 33333 23 22 0.0002 0.66 0.76 0.73 0.75

Fe NPs (Potato) 0.4878 0.9621 0.9621 0.4749 0.4709 25000 33653 33245 49595 50000 0.0003 0.0002 0.0002 0.0002 0.0002

Fe NPs (Orange) 0.6942 0.6439 0.6436 0.6955 0.6936 33333 33457 34873 34332 33333 0.0002 0.0002 0.0002 0.0002 0.0002

Fe NPs (Onion) 0.8393 0.6949 0.6949 0.8667 0.0017 25000 34233 33333 21 20 0.0002 0.0014 0.13 0.16 0.81

Fe NPs (Opuntia) 0.3581 0.2771 0.2778 0.9704 0.9478 20000 44 43 20 20 0.0006 0.0005 0.13 0.98 1.0

Fe NPs (Pomegranate) 0.964 0.9661 0.966 0.941 0.9869 58 55 58 21 22 0.18 0.16 0.18 0.64 0.62

 R2 N KF

Freundlich isotherm models Pb Cu Cr Zn Se Pb Cu Cr Zn Se Pb Cu Cr Zn Se

Fe NPs (Banana) 0.8554 0.9791 0.9792 0.9056 0.9789 5.53 5.33 6.33 7.01 7.22 9.0 9.40 9.13 8.7 6.5

Fe NPs (Potato) 0.889 0.977 0.9873 0.7134 0.7042 0.24 0.34 0.42 0.98 1.55 343 12.3 66 54 44

Fe NPs (Orange) 0.876 0.8702 0.8679 0.8701 0.8661 0.21 0.33 0.54 6.98 6.87 345 12.3 22 14 13

Fe NPs (Onion) 0.9449 0.8722 0.8679 0.9332 0.1545 0.20 6.8 7.40 7.35 7.44 388 12.3 55 14 12

Fe NPs (Opuntia) 0.9757 0.9772 0.4986 0.988 0.9767 0.64 4.4 4.46 3.65 3.98 55 1.3 1.3 1.5 1.3

Fe NPs (Pomegranate) 0.9831 0.9775 0.9725 0.9726 0.9961 0.59 1.7 1.72 1.56 1.66 66 13 11 1.4 11

3.4. Kinetic study 

Figure 13 shows the kinetics models for Cu, Pb, Se, Zn, and Cr ion adsorption onto banana, potato, orange, onion, opuntia, and pomegranate. The relation between log (qeq-qt) and t. The K1 and qe are obtained from the slope and intercept, respectively. The correlation coefficients (R2) of the pseudo-first-order kinetic model are non-linear but the pseudo-second-order kinetic model linear line, the correlation coefficients (R2) of the pseudo-second-order is shown in Table 4:. The results suggested that the overall rates of adsorption of Cu, Pb, Se, Zn, and Cr ions onto banana, potato, orange, onion, opuntia, and pomegranate were controlled by chemisorption (Ibrahim Hegazy 2021). Besides, the value of R2 (linear correlation) around 1 in pseudo-second-order confirms that the adsorption kinetics is controlled by this order and that there is a strong interaction between adsorbent and adsorbate. 

Figures 13 showed the kinetics models for Cu, Pb, Se, Zn, and Cr ion adsorption onto banana, potato, orange, onion, opuntia, and pomegranate. The pseudo-first-order kinetic model shows in Figure 13, that’s shows the relation between log (qeq-qt) and t, and through this relation the values K1 and qe could be deduced from the slope and intercept, respectively. The correlation coefficients (R2) of the pseudo-second-order kinetic model are higher than the pseudo-first-order kinetic model, and qe values calculated from the pseudo-second-order kinetic model are very close to the experimental with freundlich isotherm model. The results suggested that the overall rates of adsorption of Cu, Pb, Se, Zn, and Cr ion onto adsorption onto banana, potato, orange, onion, opuntia, and pomegranate by physical adsorption [46]. 

Figure 13: The pseudo-first-order and pseudo-second-order of Pb, Cr, Se, Cu, and Zn onto iron nanoparticles of banana, potato, orange, onion, opuntia, and pomegranate, the other conditions kept constant, the during the adsorption processes dose=04 g/L, initial concentration 3.0 mg/L, and speed 150 rpm

Table 4: The parameters of pseudo-second-order model parameters of Cu, Se, Zn, Pb, and Cr adsorption onto iron nanoparticles of banana, potato, orange, onion, opuntia, and pomegranate

Adsorbents/ metal R2 q K2

 Pb Cu Cr Zn Se Pb Cu Cr Zn Se Pb Cu Cr Zn Se

FeNPs/Potato 0.9974 0.9974 0.9974 0.9134 0.9001 10.8 10.75 7.14 7.14 7.78 0.001 0.001 0.015 0.01 0.01

FeNPs/Banana 0.9425 0.9712 0.9567 0.9223 0.9113 10.43 10.30 1.88 1.88 1.98 0.001 0.001 0.007 0.007 0.007

FeNPs/Onion 0.9325 0.9711 0.225 0.9012 0.913 9.11 9.01 8.33 8.33 8.76 0.003 0.004 0.0008 0.0008 0.0007

FeNPs/Opuntia 0.9911 0.9468 0.9911 0.915 0.9563 0.87 0.83 11.11 100 101 0.011 0.013 0.0016 0.00001 0.00001

FeNPs/Orang 0.9971 0.9717 0.9442 0.9334 0.9553 11.28 11.23 3.03 3.03 3.12 0.001 0.001 0.003 0.003 0.004

FeNPs/pomegranate 0.9974 0.9525 0.9627 0.9011 0.991 11.87 11.36 8.33 8.33 8.34 0.001 0.0015 0.005 0.003 0.003

---

## [Editor Report · Decision Letter 3]

25 Jul 2023

Adsorption of Chromium, Copper, Lead, Selenium, and Zinc Ions into Ecofriendly Synthesized Magnetic Iron Nanoparticles

PONE-D-23-05787R3

Dear Dr. El-Khateeb,

We’re pleased to inform you that your manuscript has been judged scientifically suitable for publication and will be formally accepted for publication once it meets all outstanding technical requirements.

Kind regards,

Vusumzi Pakade

Academic Editor

PLOS ONE

Additional Editor Comments (optional):

Improve the quality of the graphs.

Reviewers' comments:

<quillbot-extension-portal></quillbot-extension-portal>

---

## [Editor Report · Acceptance letter]

2 Oct 2023

PONE-D-23-05787R3 

Adsorption of Chromium, Copper, Lead, Selenium, and Zinc Ions into Ecofriendly Synthesized Magnetic Iron Nanoparticles 

Dear Dr. El-Khateeb:

I'm pleased to inform you that your manuscript has been deemed suitable for publication in PLOS ONE. Congratulations! Your manuscript is now with our production department. 

Kind regards, 

on behalf of

Prof. Vusumzi Pakade 

Academic Editor

PLOS ONE